# On the Value of Infinite Gradients in Variational Autoencoder Models

**Bin Dai**
Instiue for Advanced Study
Tsinghua University
daib09physics@hotmail.com

**Li K. Wenliang**
Gatsby Computational Neuroscience Unit
University College London
kevinli@gatsby.ucl.ac.uk

**David Wipf**
Shanghai AI Research Lab
Amazon Web Services
davidwipf@gmail.com

## Abstract

A number of recent studies of continuous variational autoencoder (VAE) models have noted, either directly or indirectly, the tendency of various parameter gradients to drift towards infinity during training. Because such gradients could potentially contribute to numerical instabilities, and are often framed as a problematic phenomena to be avoided, it may be tempting to shift to alternative energy functions that guarantee bounded gradients. But it remains an open question: What might the unintended consequences of such a restriction be? To address this issue, we examine how unbounded gradients relate to the regularization of a broad class of autoencoder-based architectures, including VAE models, as applied to data lying on or near a low-dimensional manifold (e.g., natural images). Our main finding is that, if the ultimate goal is to simultaneously avoid over-regularization (high reconstruction errors, sometimes referred to as posterior collapse) and under-regularization (excessive latent dimensions are not pruned from the model), then an autoencoder-based energy function with infinite gradients around optimal representations is provably required per a certain technical sense which we carefully detail. Given that both over- and under-regularization can directly lead to poor generated sample quality or suboptimal feature selection, this result suggests that heuristic modifications to or constraints on the VAE energy function may at times be ill-advised, and large gradients should be accommodated to the extent possible.

## 1 Introduction

Suppose we have access to continuous variables $\boldsymbol{x} \in \boldsymbol{\chi}$ that are drawn from ground-truth measure $\mu_{gt}$. This measure assigns probability mass $\mu_{gt}(d\boldsymbol{x})$ to the infinitesimal $d\boldsymbol{x}$ residing within $\boldsymbol{\chi} \subset \mathbb{R}^d$ such that we have $\int_{\boldsymbol{\chi}} \mu_{gt}(d\boldsymbol{x}) = 1$. This formalism allows us to consider data that may lie on or near an $r$-dimensional manifold embedded in $\mathbb{R}^d$ (implying $r < d$), capturing the notion of low-dimensional structure relative to the high-dimensional ambient space.

Because of the possibility of an unknown latent manifold, it is common to approximate the corresponding ground-truth measure via a density model parameterized as

$$p_\theta(\boldsymbol{x}) = \int p_\theta(\boldsymbol{x}|\boldsymbol{z})p(\boldsymbol{z})d\boldsymbol{z}. \tag{1}$$

35th Conference on Neural Information Processing Systems (NeurIPS 2021).

In this expression $\theta$ are trainable parameters and $\boldsymbol{z} \in \mathbb{R}^{\kappa}$ serves as a low-dimensional latent representation, with fixed prior $p(\boldsymbol{z}) = \mathcal{N}(\boldsymbol{0}, \boldsymbol{I})$ and ideally $\kappa \geq r$. If some $\theta^*$ were available such that $\int_A p_{\theta^*}(\boldsymbol{x}) d\boldsymbol{x} \approx \int_A \mu_{gt}(d\boldsymbol{x})$ for any measurable $A \subseteq \chi$, then the model would adequately reflect the intrinsic underlying distribution. Of course we will generally not know in advance the value of $\theta^*$, but in principle we might consider minimizing $-\log p_\theta(\boldsymbol{x})$ averaged across a set of training samples $\{\boldsymbol{x}^{(i)}\}_{i=1}^n$ drawn from $\mu_{gt}$, i.e., minimize $\frac{1}{n}\sum_{i=1}^n -\log\left[p_\theta\left(\boldsymbol{x}^{(i)}\right)\right] \approx \int -\log\left[p_\theta(\boldsymbol{x})\right]\mu_{gt}(d\boldsymbol{x})$ over $\theta$. Unfortunately though, the marginalization required to produce $p_\theta\left(\boldsymbol{x}^{(i)}\right)$ is generally intractable for models of sufficient representational power. To circumvent this issue, the variational autoencoder (VAE) [Kingma and Welling, 2014, Rezende et al., 2014] instead optimizes the tractable variational bound $\mathcal{L}(\theta, \phi) \triangleq$

$$\frac{1}{n}\sum_{i=1}^n \left\{ -\mathbb{E}_{q_\phi(\boldsymbol{z}|\boldsymbol{x}^{(i)})}\left[\log p_\theta\left(\boldsymbol{x}^{(i)}|\boldsymbol{z}\right)\right] + \mathbb{KL}\left[q_\phi(\boldsymbol{z}|\boldsymbol{x}^{(i)})||p(\boldsymbol{z})\right]\right\} \geq \frac{1}{n}\sum_{i-1}^n -\log\left[p_\theta\left(\boldsymbol{x}^{(i)}\right)\right]. \tag{2}$$

Here $q_\phi(\boldsymbol{z}|\boldsymbol{x})$ represents a variational approximation to $p_\theta(\boldsymbol{z}|\boldsymbol{x})$ with additional parameters $\phi$ governing the tightness of the bound. It is commonly referred to as an *encoder* distribution since it quantifies the mapping from $\boldsymbol{x}$ to the latent code $\boldsymbol{z}$. For analogous reasons, $p_\theta(\boldsymbol{x}|\boldsymbol{z})$ is labeled as the *decoder* distribution. When combined, the data-dependent factor $-\mathbb{E}_{q_\phi(\boldsymbol{z}|\boldsymbol{x})}\left[\log p_\theta\left(\boldsymbol{x}|\boldsymbol{z}\right)\right]$ can be viewed as instantiating a form of stochastic autoencoder (AE) structure, which attempts to assign high probability to accurate reconstructions of each $\boldsymbol{x}$; if $q_\phi(\boldsymbol{z}|\boldsymbol{x})$ is Dirac delta function, then a regular deterministic AE emerges with loss dictated by the decoder negative log-likelihood $-\log p_\theta(\boldsymbol{x}|\boldsymbol{z})$. Beyond this, $\mathbb{KL}\left[q_\phi(\boldsymbol{z}|\boldsymbol{x})||p(\boldsymbol{z})\right]$ serves as a regularization factor that pushes the encoder distribution towards the prior. The bound (2) can be minimized over $\{\theta, \phi\}$ using SGD and a simple reparameterization trick [Kingma and Welling, 2014, Rezende et al., 2014].

The latter requires that we assume a specific functional form for the encoder distribution. In this regard, it is common to select $q_\phi(\boldsymbol{z}|\boldsymbol{x}) = \mathcal{N}(\boldsymbol{z}|\boldsymbol{\mu}_z, \text{diag}[\boldsymbol{\sigma}_z]^2)$, where the Gaussian moment vectors $\boldsymbol{\mu}_z$ and $\boldsymbol{\sigma}_z$ are functions of model parameters $\phi$ and the random variable $\boldsymbol{x}$, i.e., $\boldsymbol{\mu}_z \equiv \boldsymbol{\mu}_z(\boldsymbol{x}; \phi)$, and $\boldsymbol{\sigma}_z \equiv \boldsymbol{\sigma}_z(\boldsymbol{x}; \phi)$. Similarly, for continuous data the decoder model is conventionally parameterized as $p_\theta(\boldsymbol{x}|\boldsymbol{z}) = \mathcal{N}(\boldsymbol{x}|\boldsymbol{\mu}_x, \gamma\boldsymbol{I})$, with mean defined analogously as $\boldsymbol{\mu}_x \equiv \boldsymbol{\mu}_x(\boldsymbol{z}; \theta)$ and scalar variance parameter $\gamma > 0$. The functions $\boldsymbol{\mu}_z(\boldsymbol{x}; \phi)$, $\boldsymbol{\sigma}_z(\boldsymbol{x}; \phi)$, and $\boldsymbol{\mu}_x(\boldsymbol{z}; \theta)$ are all instantiated using deep neural network layers. Given this definitions, (2) can be expressed in the more transparent form

$$\mathcal{L}(\theta, \phi) \equiv \frac{1}{n}\sum_{i=1}^n \left\{ \mathbb{E}_{q_\phi(\boldsymbol{z}|\boldsymbol{x}^{(i)})}\left[\frac{1}{\gamma}\|\boldsymbol{x}^{(i)} - \boldsymbol{\mu}_x(\boldsymbol{z}; \theta)\|_2^2\right] + d\log\gamma \tag{3}$$

$$+ \left\|\boldsymbol{\sigma}_z\left(\boldsymbol{x}^{(i)}; \phi\right)\right\|_2^2 - \log\left|\text{diag}\left[\boldsymbol{\sigma}_z\left(\boldsymbol{x}^{(i)}; \phi\right)\right]^2\right| + \left\|\boldsymbol{\mu}_z\left(\boldsymbol{x}^{(i)}; \phi\right)\right\|_2^2\right\}.$$

Although VAE models have been successfully applied to a variety of practical problems [Li and She, 2017, Schott et al., 2018, Walker et al., 2016], at times they exhibit potentially problematic behavior that has not been fully investigated. For example, a number of recent works have mentioned that if a trainable decoder variance parameter $\gamma$ is included within a VAE as in (3), then the optimal value may converge to zero resulting in infinite or unbounded gradients and potential instabilities [Dai and Wipf, 2019, Mattei and Frellsen, 2018, Rezende and Viola, 2018, Takahashi et al., 2018]. And we emphasize that this phenomena can occur *even within the confines of the stated Gaussian assumptions and inevitable regularization effects of the KL term*. While these unbounded gradients may indeed be troublesome from an optimization perspective, in this work *we will reframe such gradients as an integral part of successful autoencoder-based energy functions designed to model (in a sense that will be precisely quantified below) continuous data arising from a low-dimensional manifold.*

To accomplish this, our analysis is split into three parts. First, in Section 2 we detail how unbounded gradients contribute to an optimal, balanced form of regularization, allowing the VAE to capture low-dimensional manifold structure via a maximally parsimonious (and lossless) latent representation. Such representations turn out to be critical for tasks such as generating non-blurry samples that resemble the training data [Tolstikhin et al., 2018], or for using autoencoder-based models in general to robustly screen outliers [An and Cho, 2015, Xu et al., 2018]. Of course it is natural to consider whether these same goals could not be achieved using an alternative energy function with strictly bounded gradients.

The second and primary component of our contribution answers this question in the negative. More concretely, our main result from Section 3 proves that canonical autoencoder-based architectures will necessarily require unbounded gradients to guarantee the type of maximally parsimonious latent representation mentioned above. Thirdly, in Section 4 we elucidate the benefits of learning $\gamma$ during training, even in situations where we know that the optimal value will be at or near zero and contribute to arbitrarily-large gradients. In particular, we argue that (at the very least) learning $\gamma$ localizes troublesome unbounded gradients to narrow regions around minima of (3), while simultaneously smoothing the VAE objective across optimization trajectories prior to convergence.

Overall, our contribution can be viewed as complementary to the wide body of work analyzing what is commonly-referred to as *posterior collapse* in VAE models [He et al., 2019, Razavi et al., 2019]. The latter can be related to the situation where $\gamma$ is too large (either implicitly [Dai et al., 2020] or explicitly [Lucas et al., 2019]) and along all or most latent dimensions the posterior $q_\phi\left(\boldsymbol{z}|\boldsymbol{x}^{(i)}\right)$ collapses to the prior $\mathcal{N}(\boldsymbol{0}, \boldsymbol{I})$ leading to high reconstruction errors. In contrast, we direct our attention herein to the *opposite* condition whereby $\gamma$ is arbitrarily small and unbounded gradients invariably ensue. In this regime, the resulting latent representations obtained from bad local minimizers can potentially be under-regularized in a sense that will be described in subsequent sections.

## 2 Optimal Low-Dimensional Structure via Unbounded VAE Gradients

As alluded to previously, the VAE objective will experience unbounded gradients if $\gamma \to 0$ as has sometimes been observed (at least approximately) during training. But perhaps counter-intuitively, this phenomena nonetheless serves a critical purpose in the context of modeling data with low-dimensional manifold structure. To quantify this assertion, Section 2.1 will first precisely define what type of low-dimensional or sparse latent representations will be considered optimal for our present analysis; later in Section 2.2 we link this definition to practical VAE/AE applications.

### 2.1 Optimal Sparse (Lossless) Representations

**Definition 1** *An autoencoder-based architecture (VAE or otherwise) with decoder $\boldsymbol{\mu}_x\left(\cdot; \theta\right)$, constraint $\theta \in \Theta$, and arbitrary encoder $\boldsymbol{\mu}_z$ component[1] produces an **optimal sparse representation** of a training set $\boldsymbol{X}$ w.r.t. $\Theta$ if the following two conditions simultaneously hold:*

*(i) The reconstruction error is zero, meaning*

$$\frac{1}{n} \sum_{i=1}^{n} \left\| \boldsymbol{x}^{(i)} - \boldsymbol{\mu}_x \left[ \boldsymbol{\mu}_z \left( \boldsymbol{x}^{(i)}; \phi \right); \theta \right] \right\|_2^2 = 0. \tag{4}$$

*(ii) Conditioned on achieving perfect reconstructions per criteria (i) above, the number of latent dimensions such that $\mu_z \left( \boldsymbol{x}^{(i)}; \phi \right)_j = 0$ for all $i$ is maximal across any $\theta \in \Theta$ and any encoder function $\boldsymbol{\mu}_z$. A $j$-th latent dimension so-defined provides no benefit in reducing the reconstruction error and could in principle be removed from the model.*

**Remark 1** Conceptually, this definition is merely describing the most parsimonious latent representation of the training data (conditioned on the available capacity of the decoder) that nonetheless allows us to obtain perfect reconstructions. And when combined with the low-dimensional manifold assumption from Section 1, it readily follows that an optimal sparse representation of $\boldsymbol{X}$ will generally involve $\kappa - r$ uninformative dimensions, assuming $\kappa \geq r$ and an adequately parameterized[2] decoder family $\Theta$. As an illustrative example, for data lying on a low-dimensional linear subspace,

---

[1]The encoder $\boldsymbol{\mu}_z$ function is allowed to be unconstrained here since, unlike the decoder, it does not contribute to over-fitting (in principle even an infinite capacity encoder can be used). Additionally, the VAE encoder from (3) will also have a variance component; however, it does not directly play a role in Definition 1. We will address the relationship between $\boldsymbol{\sigma}_z$ and optimal sparsity in subsequent discussion.

[2]Obviously the decoder requires sufficient flexibility to capture the manifold structure; however, there is one additional nuance worth mentioning here. In the finite sample regime, if the decoder is allowed to be arbitrarily complex, then in principle just a single nonzero latent dimension will always be sufficient to achieve zero reconstruction error regardless of the actual data structure. This form of degenerate VAE over-fitting has been previously quantified in [Dai et al., 2018].

the corresponding optimal sparse representation obtainable via a linear decoder will be defined by the smallest subspace containing all of the data variance, i.e., the standard PCA solution.

**Remark 2** Although Definition 1 may appear to involve overly restrictive assumptions, it nonetheless well-approximates practical situations of broad interest. For example, as has been quantified in a recent study [Pope et al., 2021], natural images do indeed have a very low intrinsic dimension relative to the high-dimensional pixel space. Hence these images can in principle be reconstructed almost exactly using low-dimensional representations. Moreover, many classical under-determined inverse problems have been framed in terms of obtaining perfect reconstructions of observed measurements subject to some minimal measure of parsimony [Candès and Recht, 2009].

**Remark 3** The particular lossless notion of optimality we are adopting here is *not* meant to preclude alternatives that may be tailored for different scenarios. Rather, the proposed definition is merely selected to *showcase a class of VAE/AE usage regimes whereby infinite gradients can play an influential role*. Consequently, lossy conceptions of optimal parsimony [Alemi et al., 2016, Tishby et al., 2000], while useful in their own right, are largely outside the scope of this work.

**Remark 4** Although the encoder is generally stochastic, prior analysis from [Dai and Wipf, 2019] has revealed that the VAE global minimum is nonetheless capable of achieving something analogous to Definition 1. More concretely, for unneeded latent dimensions the posterior is pushed to the prior to optimize the KL regularizer, i.e., $q_\phi \left( z_j | \boldsymbol{x}^{(i)} \right) = \mathcal{N}(0, 1)$ for all $i$, which amounts to uninformative noise that will be filtered by the decoder so as not to impact reconstructions. In contrast, for informative dimensions the posterior variance satisfies $\sigma_z \left( \boldsymbol{x}^{(i)}; \phi \right)_j \to 0$ for all $i$. Collectively, this allows the VAE global minima to achieve

$$\frac{1}{n} \sum_{i=1}^{n} \mathbb{E}_{q_\phi(\boldsymbol{z}|\boldsymbol{x}^{(i)})} \left[ \left\| \boldsymbol{x}^{(i)} - \boldsymbol{\mu}_x \left[ \boldsymbol{z}; \theta \right] \right\|_2^2 \right] \to \frac{1}{n} \sum_{i=1}^{n} \left\| \boldsymbol{x}^{(i)} - \boldsymbol{\mu}_x \left[ \boldsymbol{\mu}_z \left( \boldsymbol{x}^{(i)}; \phi \right); \theta \right] \right\|_2^2 = 0 \quad (5)$$

while relying on the *fewest* number of active latent dimensions, such that both criteria *(i)* and *(ii)* of Definition 1 can be simultaneously satisfied.

This capability requires that the VAE avoid both over- or under-regularization of the latent representations. To be more precise, VAE *over-regularization* (sometimes loosely referred to as latent posterior collapse [He et al., 2019, Razavi et al., 2019]) occurs when too many latent dimensions are uninformative (i.e., the latent posterior along these dimensions is close to the uninformative prior) such that the reconstruction error is high and criteria *(i)* is violated. In contrast, with *under-regularized* solutions criteria *(i)* may be satisfied, and yet in reducing the reconstruction error towards zero, an excessive number of latent dimensions are informative in violation of criteria *(ii)*.

In avoiding both of these suboptimal scenarios, it can be shown that the VAE explicitly relies on $\gamma \to 0$ and the attendant unbounded gradients that follow [Dai and Wipf, 2019]. From an intuitive standpoint, we might expect that achieving criteria *(i)* would require an unbounded gradient given that, if we minimize (3) over $\gamma$ in isolation, the optimal value satisfies

$$\gamma^* = \frac{1}{dn} \sum_{i=1}^{n} \mathbb{E}_{q_\phi(\boldsymbol{z}|\boldsymbol{x}^{(i)})} \left[ \left\| \boldsymbol{x}^{(i)} - \boldsymbol{\mu}_x \left[ \boldsymbol{z}; \theta \right] \right\|_2^2 \right]. \quad (6)$$

If we then plug this value back into the $d \log \gamma$ term from (3), the result, as well as the corresponding gradients of other model parameters, is unbounded from below as the reconstruction error goes to zero (note also that in this instance, the original $1/\gamma$ data term becomes a constant, so there is no counteracting effect). Of course to actually achieve near-zero reconstruction errors, at least some dimensions of $\boldsymbol{\sigma}_z$ must be pushed towards zero as mentioned previously, which can also lead to infinite gradients within the KL-divergence factor.

## 2.2 Relevance to Typical VAE Usage Regimes

Obtaining minimalist latent representations as distilled by Definition 1 can serve a variety of practical downstream applications, such as feature extraction [Bengio et al., 2013, Ng, 2011], compression [Ballé et al., 2018, Donoho, 2006, Minnen et al., 2018], manifold learning [Silva et al., 2006], corruption removal [Dai et al., 2018], or even the generation of realistic samples. With respect to the latter, it has been shown in [Dai and Wipf, 2019] that what we have above defined as an optimal sparse representation can be viewed as a necessary (albeit not sufficient) condition for generating

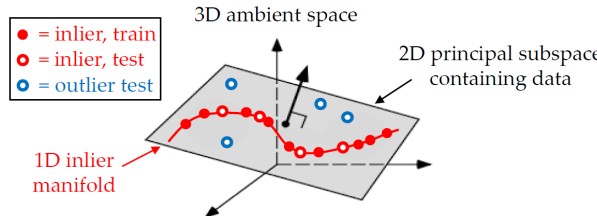

Figure 1: The importance of optimal sparse representations in screening outliers. In this example, the simple 2D principal subspace obtainable by PCA can perfectly reconstruct the inlier manifold shown in red. But this requires using two separate informative dimensions, allowing both inliers *and* outliers to be reconstructed with zero error within this subspace. In contrast, it is only by recovering the curved 1D inlier manifold, which relies on a single informative dimension, that inliers and outliers can be differentiated. Please see supplementary for practical example using real data.

samples using a continuous-space Gaussian VAE that match the training distribution. In principle, a deterministic AE architecture capable of producing optimal sparse representations can also be leveraged to generate realistic samples; this would simply involve first discarding the uninformative dimensions and then applying the same analysis from [Dai and Wipf, 2019]. In fact, variants of this strategy have been previously considered in [Ghosh et al., 2019, Tolstikhin et al., 2018].

And as a final motivational example, any AE-based architecture capable of producing optimal sparse representations can naturally be applied to screening outliers by squeezing the latent space to the minimal number of informative dimensions needed for reconstructing inliers. In doing so, we reduce the risk that outlier points $x^{(out)}$ can be accurately reconstructed by exploiting the superfluous latent flexibility. Here we are assuming that $x^{(out)} \sim \mu_{out} \neq \mu_{gt}$ for some outlier distribution $\mu_{out}$. Figure 1 contains an illustration of the basic rationale. The only exception to this line of reasoning would be adversarial outliers that follow the exact same low-dimensional structure as the inliers, meaning $\mu_{out}$ and $\mu_{gt}$ both apply all of their probability mass to the same low-dimensional manifold. In this scenario, we would need to exploit differences between $\mu_{out}$ and $\mu_{gt}$ *within* the manifold to reliably screen outliers, a regime in which Definition 1 is not directly applicable. That being said, differentiating $\mu_{out}$ and $\mu_{gt}$ once a shared low-dimensional manifold has been modeled is far easier than doing so in the original ambient space.

Additionally, in the supplementary we demonstrate that indeed, if the inlier data (in this case Fashion MNIST samples) come from a low-dimensional manifold, outlier points (MNIST samples) can be reliably differentiated, provided that $\kappa \geq r$ and the VAE has sufficient capacity and the learned $\gamma$ can converge to near zero. And because of the VAE's propensity to find optimal sparse representations where possible, even as $\kappa$ is raised such that $\kappa \gg r$, unneeded dimensions are shut off to reduce the risk of outliers masquerading as inliers (see supplementary).

### 2.3  Implications for $\beta$-VAE models

The $\beta$-VAE [Higgins et al., 2017] represents a commonly-adopted modification of the original VAE objective, whereby the KL term is rescaled by some fixed parameter $\beta > 0$. For Gaussian VAE models (which is our focus), this scale factor effectively makes no difference *if* a *fixed* decoder variance is adopted. In this situation, $\beta$ can just be directly absorbed into $\gamma$, and the $d \log \gamma$ normalization factor from (3) can be viewed as an irrelevant constant. However, if $\gamma$ is learned then $\beta \neq 1$ will make a non-trivial difference because of the imbalance introduced w.r.t. the now critical Gaussian normalization factor. In particular, if $\beta$ is too large (specifically $\beta > d/r$, where $d$ is the data dimension and $r$ is the manifold dimension), then optimal sparse representations will generally be impossible to achieve even while learning $\gamma$.

This is because, as can be inferred from the analysis in [Dai and Wipf, 2019], the VAE loss (when granted sufficient capacity) scales as $(d-r) \log \gamma$ around optimal sparse representations as $\gamma$ becomes small. In this expression, the $d \log \gamma$ factor is derived from the Gaussian normalization mentioned above, while the $-r \log \gamma$ factor originates from the KL term. However, if we scale the KL term by $\beta$ such that $\beta > d/r$, then $(d - \beta r) \log \gamma$ tends towards positive infinity as $\gamma$ becomes small, and the

VAE will instead sacrifice the reconstruction error such that optimal sparse representations are not possible.

# 3 Can we Reliably Obtain Optimal Sparse Representations without Unbounded Gradients?

As discussed in Section 2, given data originating from a low-dimensional manifold, optimal sparse representations are a necessary requirement (at least approximately) for various tasks such as generating non-blurry samples aligned with the ground-truth distribution or alternatively, screening for outliers. We have also discribed how the divergent gradients associated with $\gamma \to 0$, allow VAE global minima to achieve such optimal sparse representations. But what about alternatives that circumvent such unbounded gradients altogether? For example, could we not consider a regularized AE model that, while encouraging sparse latent representations [Ng, 2011], explicitly relies on energy function terms with bounded gradients, e.g., as may be derived from a family of sparse penalty functions with bounded gradients [Chen et al., 2017, Fan and Li, 2001, Palmer et al., 2006]? Despite this conceptual possibility, per the analysis that follows, the answer turns out to be unequivocally no within the stated context. Or more specifically, if we wish to guarantee an optimal sparse representation without additional assumptions on the decoder model and observed data, then even arbitrary AE-based objectives will necessarily require penalty terms with infinite gradients around optimal solutions.

## 3.1 A Generic AE-based Objective for Optimal Sparse Representations

Consider the constrained objective function

$$\mathcal{L}_{g,h}(\theta, \phi) \triangleq g\left(\frac{1}{dn}\sum_{i=1}^{n}\left\|\boldsymbol{x}^{(i)} - \boldsymbol{\mu}_x\left(\boldsymbol{z}^{(i)}; \theta\right)\right\|_2^2\right) + \frac{1}{d}\sum_{k=1}^{\kappa}h\left(\frac{1}{n}\|\boldsymbol{z}_k\|_2^2\right),$$

$$\text{s.t. } \boldsymbol{z}^{(i)} = \boldsymbol{\mu}_z\left(\boldsymbol{x}^{(i)}; \phi\right) \ \forall i, \ \theta \in \Theta, \tag{7}$$

where $\boldsymbol{Z} \triangleq \{\boldsymbol{z}^{(i)}\}_{i=1}^{n} \in \mathbb{R}^{\kappa \times n}$ and $\boldsymbol{z}_k$ denotes the $k$-th row of $\boldsymbol{Z}$. This expression can be viewed as characterizing a typical regularized AE with a generic penalty functions $g : \mathbb{R}^+ \to \mathbb{R}$ and $h : \mathbb{R}^+ \to \mathbb{R}$ on the reconstruction error and the norm across training samples of each latent dimension, respectively. Additionally, the constraint $\theta \in \Theta$ included in (7) can, among other things, serve to prevent the trivial solution $\boldsymbol{Z} \to \boldsymbol{0}$, which could occur if each $\boldsymbol{z}^{(i)}$ is pushed to zero while the decoder $\boldsymbol{\mu}_x$ includes an unconstrained compensatory factor that grows towards infinity such that the error $\left\|\boldsymbol{x}^{(i)} - \boldsymbol{\mu}_x\left(\boldsymbol{z}^{(i)}; \theta\right)\right\|_2$ can still be minimized to zero for all $i$. Any regularized AE must include such constraints to avoid trivial solutions, or else additional penalty terms on $\theta$ that serve a similar purpose. Note also that if we happen to choose $g = h$, the provided multipliers $1/n$, $1/d$, and $1/(dn)$ induce a form of proportional regularization within energy functions composed of multiple penalty factors of varying dimension designed to favor sparsity [Wipf and Wu, 2012]. The square-root Lasso can be viewed as a special case of this strategy that emerges when $h$ is a square-root function [Belloni et al., 2011]. However, for arbitrary selections of $g$ and $h$ (with any tunable trade-off parameter absorbed within), (7) reflects a broad family of AE architectures.

We can also relate (7) to various VAE instantiations. Define $\mathcal{I}_\infty$ as an indicator function satisfying $\mathcal{I}_\infty(u) = \infty$ for $u \neq 0$ and zero otherwise. We then have the following:

**Lemma 2** *Let* $\boldsymbol{\mu}_x\left(\boldsymbol{z}; \theta\right) = \boldsymbol{W}\boldsymbol{z} + \boldsymbol{b}$ *for some* $\boldsymbol{W} \in \mathbb{R}^{d \times \kappa}$ *and* $\boldsymbol{b} \in \mathbb{R}^d$, *and* $\boldsymbol{\sigma}_z\left(\boldsymbol{x}; \phi\right) = \boldsymbol{s}$ *for any arbitrary* $\boldsymbol{s} \in \mathbb{R}^\kappa$. *Then in the limit* $\gamma \to 0$, *the VAE loss from (3) is such that* $\min_{\boldsymbol{\sigma}_z(\boldsymbol{x}; \phi)} \mathcal{L}\left(\theta, \phi\right) \equiv \min_{\boldsymbol{s}} \mathcal{L}\left(\theta, \phi\right)$ *reduces to (7) with* $g(\cdot) = \mathcal{I}_\infty(\cdot)$ *and* $h(\cdot) = \log(\cdot)$, *excluding irrelevant constant factors.*

**Lemma 3** *For any arbitrary* $\boldsymbol{\mu}_x\left(\boldsymbol{z}; \theta\right)$ *and* $\theta \in \Theta$, *if we enforce* $\boldsymbol{\sigma}_z\left(\boldsymbol{x}; \phi\right) \to \boldsymbol{0}$ *for all* $\boldsymbol{x}$ *and apply a log transformation to each* $\|\boldsymbol{z}_k\|_2^2$, *then the VAE loss from (3) collapses to (7) with* $g(\cdot) = h(\cdot) = \log(\cdot)$, *excluding irrelevant constant factors.*

Collectively, these results point to a close affiliation between (7) and the VAE loss, especially given that $\gamma \to 0$ and $\boldsymbol{\sigma}_z\left(\boldsymbol{x}; \phi\right) \to \boldsymbol{0}$ along many dimensions are characteristics of VAE global optima [Dai and Wipf, 2019]. Hence it is natural to consider more general selections of $g$ and $h$ in the context of optimal sparse representations.

## 3.2 On the Difficulty Avoiding Unbounded Gradients

Given a generic AE architecture as in (7), this section examines what possible functions $g$ and $h$ are such that a global minimum of $\mathcal{L}_{g,h}(\theta, \phi)$ is capable of producing an optimal sparse representation. This can be addressed as follows:

**Theorem 4** *For any functions $g : \mathbb{R}^+ \to \mathbb{R}$ and $h : \mathbb{R}^+ \to \mathbb{R}$ with bounded gradients, and any dimension set $\{d, \kappa, r\}$ that order as $d \geq \kappa > r > 0$, there exists data $\boldsymbol{X} = \{\boldsymbol{x}^{(i)}\}_{i=1}^n \in \mathbb{R}^{d \times n}$ and decoder $\{\boldsymbol{\mu}_x(\boldsymbol{z}; \theta), \theta \in \Theta\}$ (with the capacity to reconstruct $\boldsymbol{x}$ lying within some parameterized family of $\kappa$-dimensional manifolds) which satisfy the following:*

*(a)* $\frac{1}{n} \sum_{i=1}^n \left\| \boldsymbol{x}^{(i)} - \boldsymbol{\mu}_x \left[ \boldsymbol{z}^{(i)}; \theta \right] \right\|_2^2 = 0$ *for some $\theta \in \Theta$ and $\boldsymbol{Z} \in \mathbb{R}^{\kappa \times n}$ with $\|\boldsymbol{z}_k\|_2 > 0$ for $r$ rows and zero elsewhere.*

*(b) Minimizing $\mathcal{L}_{g,h}(\theta, \phi)$ over $\theta$ and any possible encoder produces either a solution with $\frac{1}{n} \sum_{i=1}^n \left\| \boldsymbol{x}^{(i)} - \boldsymbol{\mu}_x \left[ \boldsymbol{z}^{(i)}; \theta \right] \right\|_2^2 > 0$ (i.e., imperfect reconstruction), or one where $\|\boldsymbol{z}_k\|_2 > 0$ for strictly more than $r$ rows of $\boldsymbol{Z}$ (i.e., not maximally sparse).*

This result effectively implies that, to guarantee every global minima corresponds with an optimal sparse reconstruction per our definition, irrespective of the decoder and observed data, the constituent penalty functions must have an unbounded gradient (at least around zero; see further intuitions below). This can be viewed as a necessary, albeit not sufficient condition, for optimal sparsity, as sufficiency requires additional care taking limits around zero, e.g., $\gamma \to 0$ in the case of the VAE. Consequently, we cannot simply replace a VAE model with a standard AE architecture to somehow guarantee optimal sparse representations devoid of infinite surrounding gradients (unless further assumptions on the data and decoder are introduced).

## 3.3 High-Level Intuition Behind Theorem 4

While the proof is predicated on a nuanced counterexample designed with a specific technical purpose in mind (see supplementary file), we can nonetheless loosely convey the basic idea through a toy illustration shown in Figure 2. Here we are assuming that the data points $\{\boldsymbol{x}^{(i)}\}_{i=1}^n$ lie on a 1D manifold embedded in 2D ambient space; the extension to higher dimensions is straightforward. Moreover, we stipulate that this manifold is tightly squeezed within a small non-negative $\epsilon \times \epsilon$ square near zero, represented by the blue curve on the lefthand side of Figure 2. Now consider a sample point $\boldsymbol{x}' = [x_1', x_2']^\top$ taken from somewhere along the stated 1D manifold. We represent this point using two candidate decoder functions, both assumed to be within the capacity of $\boldsymbol{\mu}_x$, as displayed in the middle of Figure 2.

For the simple decoder case, which is just the identity function $\boldsymbol{\mu}_x(\boldsymbol{z}; \theta) = \boldsymbol{z}$, the values of $z_1 = z_1'$ and $z_2 = z_2'$ needed for a perfect reconstruction will both be small, i.e., $\{z_1', z_2'\} \leq \epsilon$ by design. In contrast, the optimal decoder only requires that a single dimension of $\boldsymbol{z}$, namely $z_1$, be nonzero. However, the optimal value actually needed for perfect reconstruction, denoted $z_1^*$, can be arbitrarily large in controlling where along the extended, labyrinthine manifold pathway $\boldsymbol{x}'$ is located (for ease of presentation we will assume $z_1^*$ is also positive). Hence we can easily have that

$$z_1^* \gg \epsilon \geq \max(z_1', z_2'). \tag{8}$$

Because of this, to ensure that $\boldsymbol{z}^* = [z_1^*, 0]^\top$ is preferred over the $\boldsymbol{z}'$ alternative, we require a concave penalty function $h$ on each encoder dimension such that any infinitesimal movement away from zero incurs an arbitrarily-large cost, while increases originating from points away from zero incur only a modest additional cost (see the green curve on the righthand side of Figure 2). From this it follows that any movement of $z_1'$ and $z_2'$ away from zero, no matter how small, will be such that we can guarantee that the penalties on $\boldsymbol{z}^*$ and $\boldsymbol{z}'$ will satisfy

$$h(z_1^*) + h(0) = h(z_1^*) \approx h(z_1') \approx h(z_2') < h(z_1') + h(z_2') \approx 2[h(z_1^*) + h(0)], \tag{9}$$

and so $\boldsymbol{z}_*$ is preferred. The righthand side of Figure 2 motivates this relationship. Note also that if we were to explicitly bound the slope of $h$ around zero, then we could always select an $\epsilon$ sufficiently small such that the inequality in (9) is reversed; hence an unbounded slope is required to achieve the stated result.

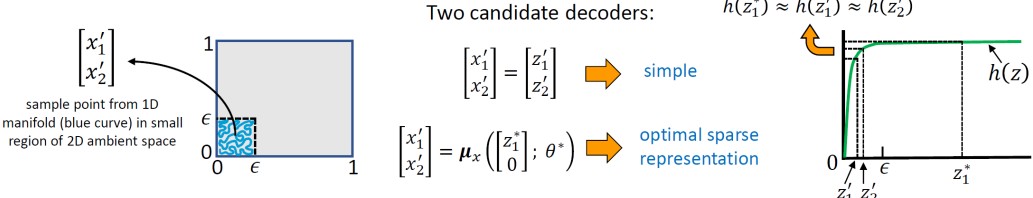

Figure 2: 2D illustration of the intuition behind Theorem 4. See Section 3.3 for details.

To a large extent, the intuition here mirrors the basic scenario from Figure 1, and is emblematic of broader situations that naturally arise in practice. For example, if we run PCA on MNIST data, we find that only a 100 or so principal components are needed to achieve highly accurate reconstructions. But a VAE model with only around 15 informative latent dimensions can accomplish something similar [Dai et al., 2018] by closely approximating an optimal sparse representation using a nonlinear decoder. Of course unless we have an objective function with a strong preference for lower-dimensional structures, as instantiated through large gradients around optimal sparse representations, then the network may well favor or converge to a simpler, higher-dimensional alternative (e.g., resembling a PCA solution).

## 4 Mitigating Unbounded Gradients via $\gamma$-Dependent Smoothing

While we have argued that unbounded gradients may serve a useful purpose in obtaining optimal latent representations, they may nonetheless pose challenges from an optimization standpoint. In addressing this concern, it is worth acknowledging that energy functions involving infinite gradients and/or unbounded regions are already indispensable across a wide range of structured regression and sparse estimation problems [Gorodnitsky and Rao, 1997, Rao et al., 2003]. This history implies that when training a VAE or other related AE model, we may borrow appropriate tools designed to mitigate the risk of converging to bad local solutions or regions of instability. In this vein, one effective strategy involves partially minimizing what amounts to a smoothed version of the original objective function. The degree of smoothness is then gradually reduced as the optimization trajectory moves towards an optimum. Within the domain of underdetermined linear inverse problems, this procedure is frequently used to find maximally sparse representations with minimal reconstruction error [Chartrand and Yin, 2008, Hu et al., 2012, Xu et al., 2013].

The VAE automatically accomplishes something similar when we choose to iteratively estimate $\gamma$ during training rather than merely setting its value to near zero as may be theoretically optimal (assuming we know that there exists sufficient network capacity to achieve negligible reconstruction errors). Initially, when the reconstruction cost is still high because encoder/decoder parameters have not converged, the learned $\gamma$ will be larger and the overall VAE energy will be relatively smooth, devoid of many deep local minimizers. It is only later as the data fit $\sum_{i=1}^{n} \mathbb{E}_{q_\phi(\boldsymbol{z}|\boldsymbol{x}^{(i)})} \left[ \left\| \boldsymbol{x}^{(i)} - \boldsymbol{\mu}_x(\boldsymbol{z};\theta) \right\|_2^2 \right]$ becomes small that $\gamma$ will follow suite, and by this point it is more likely that we have already approached a basin of attraction capable of producing optimal sparse reconstructions. Additionally, unlike fixing $\gamma \approx 0$ for all training iterations, in which case gradients will be unbounded right from the beginning, by learning $\gamma$ we will likely only encounter large gradients in a narrow neighborhood around minimizing solutions. This implies that in practice, we only need accommodate such gradients when the reconstruction error becomes small, at which point stability countermeasures can be deployed if/when necessary, e.g., reduced step size, checks for oscillating gradient sign patterns [Riedmiller and Braun, 1993], etc.

To help visualize these points, in Figure 3 we have plotted 1D slices through the objective function of a simple VAE model involving a single layer for both encoder and decoder, applied to data from a random low-dimensional subspace. We vary $\gamma \in \{10^{-3}, 10^{-2}, 10^{-1}, 1\}$, which exposes the increasing gradients and multi-modal nature of the objective function as $\gamma$ becomes smaller. Dashed vertical lines indicate the minimal value of the respective curve for each $\gamma$. Additionally, we have explicitly designed the model underpinning this visualization such that there will exist an optimal sparse representation at zero on the $x$-axis. Consequently, we can readily observe that as $\gamma$ becomes

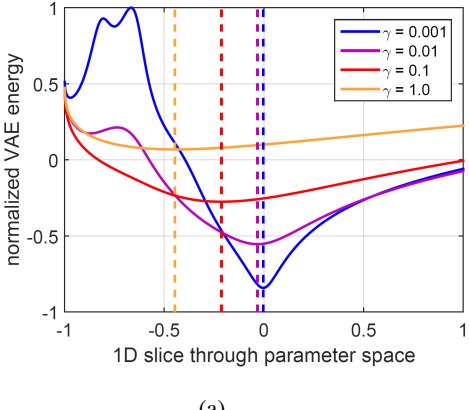
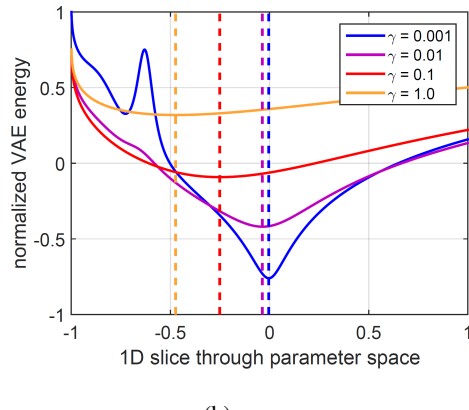

(a)                                               (b)

Figure 3: Plots (a) and (b) show two sets of representative 1D slices through the VAE objective function (3) as the value of $\gamma$ is varied. Dashed vertical lines indicate the $x$-axis location of the minimal value of each respective slice and $\gamma$ setting. And for both plots (a) and (b) the 1D slices are set such that an optimal sparse representation would occur at zero on the $x$-axis when $\gamma \to 0$. It can be observed that disconnected local minima only occur when $\gamma$ is small.

sufficiently small, the minimizing value of the VAE energy increasingly aligns with an optimal sparse representation as desired. However, as $\gamma$ is reduced the energy is less smooth and disconnected local minima appear in both 1D slices. And local minima of the VAE loss surface can at times be risk points for under-regularized representations.

To further explore the implications of this $\gamma$-dependent smoothing effect, we empirically compare a practical scenario whereby learning $\gamma$ may be better than fixing it to an arbitrarily small value. To this effect, we first train a VAE model on CelebA data [Liu et al., 2015] and learn an appropriate small value of $\gamma$ denoted $\gamma^*$ (note that $\gamma^*$ need not be exactly zero since with real data and limited capacity the network will generally display some nonzero reconstruction errors). Please see the supplementary for network and training details. We then retrain the same network from scratch but with $\gamma = \gamma^*$ fixed throughout all training iterations.

The resulting models are evaluated via the reconstruction error and the maximum mean discrepancy (MMD) between the aggregated posterior $q_\phi(z) \triangleq \frac{1}{n} \sum_i q_\phi(z|x^{(i)})$ [Makhzani et al., 2016] and the prior $p(z) = \mathcal{N}(0, I)$. If too few latent dimensions are removed by swamping the appropriate channels with noise following the prior (i.e., under-regularization), then we would expect $q_\phi(z)$ to be confined near a low-dimensional manifold in $\mathbb{R}^\kappa$ and the MMD to be much larger. Note that for ideal generative modeling performance via an autoencoder architecture, it is required that

$$\frac{1}{n} \sum_i q_\phi(z|x^{(i)}) \approx \int_{\mathcal{X}} q_\phi(z|x)\mu_{gt}(dx) = p(z), \tag{10}$$

meaning the MMD from $\mathcal{N}(0, I)$ is ideally zero [Makhzani et al., 2016]. With manifold data this is only possible if an optimal sparse representation is produced by the VAE or autoencoder-based analogue [Tolstikhin et al., 2018].

Results are displayed in Figure 4(a), where as expected the reconstruction errors are nearly identical, but the learnable $\gamma$ case leads to much lower MMD values, indicative of a better local solution with reduced under-regularization. We also plot the evolution of the gradient magnitudes $\left\| \frac{d\mathcal{L}(\theta,\phi)}{dz} \right\|_2$ in Figure 4(b) (other gradients are similar). When $\gamma$ is learned, the gradient increases slowly; however, with fixed $\gamma = \gamma^*$, there exists a large gradient right from the start since $\gamma^*$ is small but the reconstruction error is high. This contributes to a worse final solution per the results in Figure 4(a). Additionally, examples of using a learnable $\gamma$ to improve generated sample quality based on these principles can be found in [Dai and Wipf, 2019].

We close this section with one notable caveat: Although learning $\gamma$ can be beneficial for the reasons we have given, it is not a panacea and in certain situations there can be unintended consequences. For

|  | CelebA | |
|---|---|---|
|  | Rec. Err. | MMD |
| Learnable $\gamma$ | 352.8 | 93.3 |
| Fix $\gamma = \gamma^*$ | 349.9 | 291.8 |

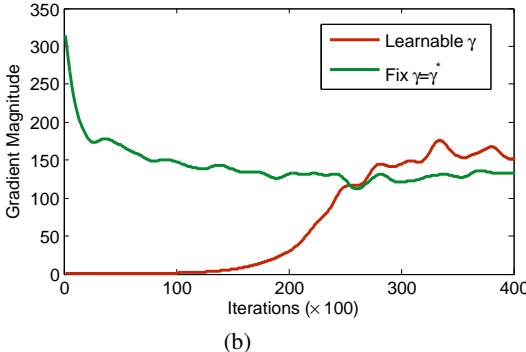

(a)  (b)

Figure 4: (a) Reconstruction error and MMD between $q_\phi(z)$ and $\mathcal{N}(0, I)$ on CelebA ($128 \times 128$ resolution). We first train a VAE with learnable $\gamma$ and obtain the optimal value $\gamma^*$. Then we fix $\gamma = \gamma^*$ and re-train the same network from scratch. While the final reconstruction errors are almost the same, the MMDs between $q_\phi(z)$ and the prior $\mathcal{N}(0, I)$ are significantly different. (b) The Evolution of the gradient $\left\| \frac{d\mathcal{L}(\theta, \phi)}{dz} \right\|_2$. Although both curves end up with similar final values, the large initial gradient with fixed $\gamma$ is disruptive to the final solution.

example, if a particular VAE model experiences posterior collapse during training, then it may be necessary to place an upper bound on $\gamma$ to help reduce the collapse risk.

## 5   Conclusion

It is not uncommon to learn the VAE decoder variance parameter in situations where the training data has a noise component that we are unable or do not wish to model. By doing so we can avoid tuning a trade-off parameter while allowing the model to adapt to the data. However, with sufficient capacity networks and relatively clean data, the risk of unbounded gradients when training $\gamma$ has frequently been raised as a potentially problematic phenomena. We nonetheless provide formal justification for this choice (even in cases where $\gamma$ does tend to zero) on two primary fronts:

- We prove that unbounded gradients are in fact necessary for guaranteeing that global minima of canonical AE architectures will coincide with optimal sparse representations, meaning high fidelity reconstruction of the training data using the minimal number of informative latent dimensions. Hence there is no obvious alternative if this form of parsimony is our goal. Furthermore, given the value of such representations to numerous downstream tasks as described in Section 2.2, our analysis suggests that heuristic modifications to or constraints on the VAE energy function may be ill-advised, and large gradients should be accommodated to the extent possible (e.g., reduced step size, checks for oscillating gradient sign patterns, etc.).

- We present compelling evidence that by learning $\gamma$, large gradients away from global minimizers, as well as at least some bad local minimizers, can be mitigated or smoothed within the VAE loss surface. This helps to explain observed successes learning $\gamma$ in situations where the optimal value turns out to be small or near zero. Note that as mentioned in Section 1, it is already known that fixing $\gamma$ too *high* can lead to over-regularization and the widely-studied phenomena of posterior collapse [He et al., 2019, Lucas et al., 2019, Razavi et al., 2019]. In a similar vein, we have demonstrated the complementary yet underappreciated fact that prematurely fixing $\gamma$ too *low*, even to what may ultimately be the optimal value near zero, can steer convergence towards under-regularized local minima and the inadvertent wasteful deployment of latent degrees-of-freedom.

And finally, although not our focus, our results herein naturally relate to more flexible VAE models with non-Gaussian latent posteriors [Kingma et al., 2016, Rezende and Mohamed, 2015] or adaptable/trainable priors [Bauer and Mnih, 2019, Tomczak and Welling, 2018]. While these types of enhancements can be useful tools for favoring $q_\phi(z) \approx p(z)$, they do not circumvent the infinite gradients that will occur around optimal sparse representations.

## Funding Transparency Statement

Some initial components of this work were conceived while Bin Dai was an intern and David Wipf was an FTE at Microsoft Research in Beijing. Additionally, Li K. Wenliang contributed as an intern and David Wipf as an FTE at the AWS Shanghai AI Research Lab. There are no other sources of funding to report.

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
