# SUPPLEMENTARY FILE:
# On the Value of Infinite Gradients in Variational Autoencoder Models

**Bin Dai**
Institue for Advanced Study
Tsinghua University
daib09physics@hotmail.com

**Li K. Wenliang**
Gatsby Computational Neuroscience Unit
University College London
kevinli@gatsby.ucl.ac.uk

**David Wipf**
Shanghai AI Research Lab
Amazon Web Services
davidwipf@gmail.com

## 6   Outlier Detection Example

As mentioned in Section 2.2 of the main text, the ability of the VAE to produce optimal sparse representations can potentially be leveraged to detect outliers. To test this possibility, we employ the basic experimental paradigm from [Nalisnick et al., 2019], training models on a given inlier set and then comparing evaluation metrics applied to both inlier train/test samples and distinct outlier samples. Consistent with [Nalisnick et al., 2019] and convention elsewhere, we use the bits-per-dimension (BPD) metric for expressing aggregate results in a convenient range for tables. This metric correlates with the negative log-likelihood, with lower values indicating a higher likelihood. Please see [Theis et al., 2016] for further information regarding this metric and how it is computed. In general, a successful model should produce *higher* BPD scores for the outlier samples than the inlier test samples.

Within this setup, we examine the ability of VAE models with a learned $\gamma$ to differentiate inlier and outlier distributions as $\kappa$, the VAE latent space dimension, is varied. Because of the VAE's preference for sparse latent representations, we expect that increasing $\kappa$ will not inadvertently allow outliers to be assigned accurate reconstructions and ultimately lower BPD scores. FashionMNIST [Xiao et al., 2017] images are chosen as the inlier distribution while MNIST [LeCun et al., 1998] serves as the outlier distribution. As reported in [Nalisnick et al., 2019] and included in Table 1 below, a flow-based Glow generative model [Kingma and Dhariwal, 2018] trained on FashionMNIST produced lower BPD scores on MNIST, indicating that outlier data was errantly preferred.

In contrast, when we trained our VAE baseline with a learned $\gamma$ (see architecture specifics in the next section below), Table 1 results reveal that FashionMNIST has clearly lower BPD scores across $\kappa$ values ranging from 16 to 512. This indicates that the model has in some sense correctly rejected the outliers independently of $\kappa$ as would be expected for a model capable of finding an optimal sparse representation. Note also that the BPD values for the test data stabilize such that even with what would seem to be excessively high $\kappa$ values (e.g., $\kappa = 512$, which is larger than the number of nonzero pixels in typical MNIST samples), the extra degrees-of-freedom do not provide an inadvertent pathway for the outlier samples to receive undesirable acceptance.

The learned $\gamma$ is also reported in the last row of Table 1. We can observe that these values remain small and stable implying that additional latent degrees-of-freedom are not needed to improve the inlier data fit. Conversely, if the learned $\gamma$ values were to display a consistently strong inverse correlation across the $\kappa$ range, it would indicate that extra latent dimensions were actually required

35th Conference on Neural Information Processing Systems (NeurIPS 2021).

| | Glow | 16 | 32 | 64 | 128 | 256 | 512 |
|---|---|---|---|---|---|---|---|
| FashionMNIST-Train | 2.902 | 2.375 | 2.208 | 2.084 | 2.009 | 2.054 | 2.020 |
| FashionMNIST-Test | 2.958 | 2.805 | 2.690 | 2.562 | 2.424 | 2.397 | 2.301 |
| MNIST-Test | 1.833 | 9.598 | 8.618 | 6.294 | 4.958 | 4.578 | 4.351 |
| $\gamma$ | – | 0.0055 | 0.0040 | 0.0030 | 0.0024 | 0.0023 | 0.0024 |

Table 1: BPD values for VAE models trained on FashionMNIST as $\kappa$ is varied from 16 to 512. When $\kappa$ increases, the BPD saturates while robustly differentiating inliers and outliers.

for fitting the inlier data, and that a low-dimensional inlier manifold may not have been found. This would be the case, for example, if the VAE architecture did not have sufficient capacity to adequately model the inlier manifold.

# 7 Network and Training Details

This section provides the network and training details related to both the gradient magnitude and MMD tests from Figure 4 in the main paper, as well as the outlier experiments described above.

**Gradient Magnitude and MMD Experiments:** The input CelebA images were resized to $128 \times 128$ via center cropping (and down-sampling as well for the $64 \times 64$ tests). As for the VAE architecture, the encoder consists of 5 convolution layers, each followed by a batch normalization layer and a ReLU activation layer. The kernel size and the stride are 5 and 2 for all the convolution layers. The number of channels is 64, 128, 256, 512, and 1024 respectively. The size of the output feature map is $4 \times 4 \times 1024$. We then use a $1 \times 1$ convolution layer to transform it to $4 \times 4 \times 4$. This feature map is then flattened and fed to two fully connected layers with 64 dimensions corresponding to $\boldsymbol{\mu}_z$ and $\log \boldsymbol{\sigma}_z$. The decoder reverses the structure of the encoder. The network is optimized using Adam optimizer for $40K$ iterations with learning rate equal to $0.0001$.

**Outlier Experiments:** Input FashionMNIST and MNIST images are $32 \times 32$. For the VAE model, the encoder consists of 3 convolution layers, each followed by a batch normalization layer and ReLU activation layer. The kernel size and the stride are 5 and 2 for all the convolution layers. The number of channels is 64, 128, and 256 respectively. The output feature map is then flattened to a $\kappa$-dimensional feature vector. We use two 64-dimensional fully connected layers to produce $\boldsymbol{\mu}_z$ and $\log \boldsymbol{\sigma}_z$ given the feature vector. The decoder just reverses the structure of the encoder. The network is optimized using Adam optimizer for $200K$ iterations with learning rate equal to $0.0001$.

# 8 Proof of Technical Results

This section includes proofs for Lemma 2, Lemma 3, and Theorem 4 from the main text.

## 8.1 Proof of Lemma 2

Given the stated assumptions, we may conclude that after minimizing over $\boldsymbol{\sigma}_z(\boldsymbol{x}; \phi)$, the resulting VAE loss reduces to

$$\mathcal{L}(\theta, \phi) \equiv \frac{1}{n} \sum_{i=1}^{n} \left\{ \frac{1}{\gamma} \left\| \boldsymbol{x}^{(i)} - \boldsymbol{W} \boldsymbol{\mu}_z^{(i)} - \boldsymbol{b} \right\|_2^2 + \sum_{k=1}^{\kappa} \log \left( \gamma + \|\boldsymbol{w}^{(k)}\|_2^2 \right) + \left\| \boldsymbol{\mu}_z^{(i)} \right\|_2^2 + (d - \kappa) \log \gamma \right\},$$

(11)

where $\boldsymbol{w}^{(k)}$ denotes the $k$-th column of $\boldsymbol{W}$ and for notational simplicity we adopt $\boldsymbol{\mu}_z^{(i)} \equiv \boldsymbol{\mu}_z \left( \boldsymbol{x}^{(i)}; \phi \right)$ (see the proof from [Dai et al., 2018][Lemma 1] for the details of producing (11)). From this expression we note that

$$\lim_{\gamma \to 0} \frac{1}{\gamma} \left\| \boldsymbol{x}^{(i)} - \boldsymbol{W} \boldsymbol{\mu}_z^{(i)} - \boldsymbol{b} \right\|_2^2 = \mathcal{I}_{\infty} \left[ \boldsymbol{x}^{(i)} = \boldsymbol{W} \boldsymbol{\mu}_z^{(i)} + \boldsymbol{b} \right],$$

(12)

where the indicator $\mathcal{I}_\infty$ satisfies $\mathcal{I}_\infty(u) = \infty$ for $u \neq 0$ and zero otherwise. Now for the moment assume that $d = \kappa$ such that the last term in (11) is zero. Then in the limit $\gamma \to 0$, (11) becomes

$$
\begin{aligned}
\mathcal{L}(\theta, \phi) &= \frac{1}{n}\sum_{i=1}^{n}\left\{\mathcal{I}_\infty\left[\boldsymbol{x}^{(i)} = \boldsymbol{W}\boldsymbol{\mu}_z^{(i)} + \boldsymbol{b}\right] + \sum_{k=1}^{\kappa}\log\left(\|\boldsymbol{w}^{(k)}\|_2^2\right) + \left\|\boldsymbol{\mu}_z^{(i)}\right\|_2^2\right\} \\
&\equiv \frac{1}{n}\sum_{i=1}^{n}\left\{\sum_{k=1}^{\kappa}\log\left(\|\boldsymbol{w}^{(k)}\|_2^2\right) + \left\|\boldsymbol{\mu}_z^{(i)}\right\|_2^2\right\} \quad \text{s.t. } \boldsymbol{x}^{(i)} = \boldsymbol{W}\boldsymbol{\mu}_z^{(i)} + \boldsymbol{b}, \ \forall i. \quad (13)
\end{aligned}
$$

At this point, let $\widetilde{\boldsymbol{W}}$ denote $\boldsymbol{W}$ with columns set to unit $\ell_2$-norm, $\boldsymbol{\Omega} \in \mathbb{R}^{\kappa \times \kappa}$ indicate a diagonal matrix with $k$-th diagonal element $\omega_k = \|\boldsymbol{w}^{(k)}\|_2^2$ for all $k = 1, \ldots, \kappa$, and $\boldsymbol{Z} = \{\boldsymbol{z}^{(1)}, \ldots, \boldsymbol{z}^{(n)}\} \in \mathbb{R}^{\kappa \times n}$ with $\boldsymbol{z}^{(i)} = \boldsymbol{\Omega}\boldsymbol{\mu}_z^{(i)}$. This implies that (13) can be reparameterized as

$$
\begin{aligned}
\mathcal{L}(\theta, \phi) &= \frac{1}{n}\sum_{i=1}^{n}\left\{\sum_{k=1}^{\kappa}\log\left(\omega_k^2\right) + \frac{(z_k^{(i)})^2}{\omega_k^2}\right\} \quad \text{s.t. } \boldsymbol{x}^{(i)} = \widetilde{\boldsymbol{W}}\boldsymbol{z}^{(i)} + \boldsymbol{b}, \ \forall i \\
&\equiv \sum_{k=1}^{\kappa}\left(\log\left(\omega_k^2\right) + \frac{\frac{1}{n}\|\boldsymbol{z}_k\|_2^2}{\omega_k^2}\right) \quad \text{s.t. } \boldsymbol{x}^{(i)} = \widetilde{\boldsymbol{W}}\boldsymbol{z}^{(i)} + \boldsymbol{b}, \ \forall i, \quad (14)
\end{aligned}
$$

where $\boldsymbol{z}_k$ is the $k$-th row of $\boldsymbol{Z}$. If we then minimize across $\omega_k$ for all $k$, we find that the optimal solution satisfies $(\omega_k^*)^2 = \frac{1}{n}\|\boldsymbol{z}_k\|_2^2$. When we plug this value into (14) and exclude constant terms, we arrive at

$$
\mathcal{L}(\theta, \phi) \equiv \sum_{k=1}^{\kappa}\log\left(\tfrac{1}{n}\|\boldsymbol{z}_k\|_2^2\right) \quad \text{s.t. } \begin{aligned} \boldsymbol{x}^{(i)} &= \boldsymbol{W}\boldsymbol{z}^{(i)} + \boldsymbol{b} \ \forall i, \\ \boldsymbol{z}^{(i)} &= \boldsymbol{\mu}_z\left(\boldsymbol{x}^{(i)}; \phi\right) \ \forall i, \\ \theta &\in \Theta, \end{aligned} \quad (15)
$$

where the column norm constraint on $\boldsymbol{W}$ is now enforced by the constraint set $\Theta$ and the rescaling by $\boldsymbol{\Omega}$ has been absorbed into $\boldsymbol{\mu}_z^{(i)} = \boldsymbol{\mu}_z\left(\boldsymbol{x}^{(i)}; \phi\right)$. Hence we have explicitly converted the VAE loss into a special case of $calL_{g,h}(\theta, \phi)$ when we assume that $g(\cdot) = \mathcal{I}_\infty(\cdot)$ and $h(\cdot) = \log(\cdot)$, inconsequential reparameterizations and constants notwithstanding.

And finally, even if $d \neq \kappa$, the same derivations can proceed as before. Note that adding a fixed constant, even one that can grow arbitrarily large or small, does not alter the effective loss. Moreover, regardless of the $(d - \kappa)\log\gamma$ term, because $\lim_{\gamma \to 0^+} \frac{a}{\gamma} + \log(b + \gamma) + c\log\gamma = \infty$ for any $a > 0$, $b \geq 0$, and $c \in \mathbb{R}$, the constraint $a = 0$ (or more precisely $\boldsymbol{x}^{(i)} = \boldsymbol{W}\boldsymbol{\mu}_z^{(i)} + \boldsymbol{b}$ for all $i$) must be enforced at any feasible limiting solution. $\blacksquare$

## 8.2 Proof of Lemma 3

As discussed in Section 2.1, value of $\gamma$ that minimizes (3), all else being fixed, satisfies

$$
\gamma^* = \frac{1}{dn}\sum_{i=1}^{n}\mathbb{E}_{q_\phi(\boldsymbol{z}|\boldsymbol{x}^{(i)})}\left[\left\|\boldsymbol{x}^{(i)} - \boldsymbol{\mu}_x\left[\boldsymbol{z}; \theta\right]\right\|_2^2\right]. \quad (16)
$$

If we plug this value into (3) and remove irrelevant constant factors and terms that only depend on $\boldsymbol{\sigma}_z(\boldsymbol{x}; \phi)$, we arrive at the simplified loss

$$
\mathcal{L}(\theta, \phi) = d\log\left(\frac{1}{dn}\sum_{i=1}^{n}\mathbb{E}_{q_\phi(\boldsymbol{z}|\boldsymbol{x}^{(i)})}\left[\left\|\boldsymbol{x}^{(i)} - \boldsymbol{\mu}_x\left[\boldsymbol{z}; \theta\right]\right\|_2^2\right]\right) + \frac{1}{n}\sum_{i=1}^{n}\left\{\left\|\boldsymbol{\mu}_z\left(\boldsymbol{x}^{(i)}; \phi\right)\right\|_2^2\right\}. \quad (17)
$$

We next enforce the limit $\boldsymbol{\sigma}_z(\boldsymbol{x}; \phi) \to \boldsymbol{0}$ for all $\boldsymbol{x}$ such that (17) further reduces to

$$
\begin{aligned}
\mathcal{L}(\theta, \phi) &= d\log\left(\frac{1}{dn}\sum_{i=1}^{n}\left\|\boldsymbol{x}^{(i)} - \boldsymbol{\mu}_x\left[\boldsymbol{z}^{(i)}; \theta\right]\right\|_2^2\right) + \frac{1}{n}\sum_{i=1}^{n}\left\{\left\|\boldsymbol{z}^{(i)}\right\|_2^2\right\} \\
&\equiv d\log\left(\frac{1}{dn}\sum_{i=1}^{n}\left\|\boldsymbol{x}^{(i)} - \boldsymbol{\mu}_x\left[\boldsymbol{z}^{(i)}; \theta\right]\right\|_2^2\right) + \sum_{k=1}^{\kappa}\left\{\frac{1}{n}\|\boldsymbol{z}_k\|_2^2\right\}, \quad (18)
\end{aligned}
$$

where $\boldsymbol{z}^{(i)} \triangleq \boldsymbol{\mu}_z \left( \boldsymbol{x}^{(i)}; \phi \right)$, $\boldsymbol{Z} = \{ \boldsymbol{z}^{(1)}, \dots, \boldsymbol{z}^{(n)} \} \in \mathbb{R}^{\kappa \times n}$, and $\boldsymbol{z}_k$ denotes the $k$-th row of $\boldsymbol{Z}$. We may then apply the stated log transformation to each $\| \boldsymbol{z}_k \|_2^2$ (or more precisely $\frac{1}{n} \| \boldsymbol{z}_k \|_2^2$) and divide both sides by $d$, an inconsequential rescaling. This produces the final loss

$$\mathcal{L}(\theta, \phi) = \log \left( \frac{1}{dn} \sum_{i=1}^{n} \left\| \boldsymbol{x}^{(i)} - \boldsymbol{\mu}_x \left[ \boldsymbol{z}^{(i)}; \theta \right] \right\|_2^2 \right) + \frac{1}{d} \sum_{k=1}^{\kappa} \left\{ \log \left( \frac{1}{n} \| \boldsymbol{z}_k \|_2^2 \right) \right\}, \qquad (19)$$

which is exactly the form of $\mathcal{L}_{g,h}(\theta, \phi)$ with $g(\cdot) = h(\cdot) = \log(\cdot)$. $\blacksquare$

## 8.3 Proof of Theorem 4

To begin, we assume that $h(u)$ is a concave, non-decreasing function defined on the domain $u \geq 0$. These are central characteristics of sparsity inducing penalty functions [Chen et al., 2017, Palmer et al., 2006] and it is not difficult to show that additional flexibility does not gain us anything in the present context. For convenience, we assume that $h$ is differentiable everywhere, although this condition can also be relaxed. We then focus on the case where the gradient of $h$ is bounded. Per these specifications, the largest gradient will necessarily occur at $h'(0) \equiv \lim_{u \to 0^+} h'(u)$. Note also that this limiting gradient cannot equal zero; otherwise we trivially default to a flat penalty function such that all solutions have equal cost and the theorem guarantee is unattainable right from the start.

From here, the basic idea is to construct a counterexample that satisfies the conditions of the theorem, and yet involves a simple network structure that, if $h'(u)$ is bounded around zero, is unable to minimize the stated objective using at most $r$ nonzero rows of $\boldsymbol{Z}$ while simultaneously achieving zero reconstruction error. To this end, we first consider a low-dimensional example that satisfies the constraints $d \geq \kappa > r > 0$. Later we discuss the trivial generalization to higher dimensions.

Consider the two-dimensional latent representation $\boldsymbol{z} = [z_1, z_2]^\top$ and a single-parameter decoder that computes

$$\boldsymbol{\mu}_x \left( \boldsymbol{z}; \theta \right) = \theta \pi \left( t \left[ z_1 \right] \right) + (1 - \theta) \begin{bmatrix} z_1 \\ z_2 \end{bmatrix}, \qquad (20)$$

where $\theta \in \Theta \triangleq [0, 1]$ is a scalar parameter, $t : \mathbb{R} \to (0, 1)$ squashes its argument to the interval between zero and one (e.g., a logistic function), and $\pi : [0, 1] \to \mathcal{S} \subset [0, 1]^2$ is for now an arbitrary function defined on the stated interval, mapping to a subset $\mathcal{S}$ of the unit square $[0, 1]^2$. Per this construction, the decoder can be viewed as a tunable mixture weighted by $\theta$, and for either $\theta = 0$ or $\theta = 1$, the range of the decoder $\boldsymbol{\mu}_x \left( \boldsymbol{z}; \theta \right)$ is contained within $[0, 1]^2$.

Now suppose we have training samples $\{ \boldsymbol{x}^{(i)} \}_{i=1}^n$ that were produced via the generative process

$$z_{gt}^{(i)} \sim p\left( z_{gt} \right) \quad \text{and} \quad \boldsymbol{x}^{(i)} = \pi \left( t \left[ z_{gt}^{(i)} \right] \right) \qquad (21)$$

for some prior $p\left( z_{gt} \right)$ on the ground-truth latent variable $z_{gt} \in \mathbb{R}$. Furthermore, assume that the function $\pi$ is such that for all $t \left[ z_{gt} \right] \in [C, 1]$ with constant $C < 1$, $\begin{bmatrix} x_1 \\ x_2 \end{bmatrix} = \pi \left( t \left[ z_{gt} \right] \right)$ satisfies $0 < |x_j| < \epsilon$ for $j = 1, 2$, with $\epsilon > 0$ arbitrarily small. We also stipulate that $p\left( z_{gt} \right)$ places all (or almost all) of its probability mass such that $t \left[ z_{gt} \right] \in [C, 1]$, which implies that the observed training points will all be arbitrarily close to zero.

Given this observed data, we can then evaluate the optimal AE for different penalties $h$. We allow that the encoder is sufficiently complex such that

$$\min_{\phi} \mathcal{L}_{g,h}(\theta, \phi) \equiv \min_{\boldsymbol{Z}} g \left( \frac{1}{dn} \sum_{i=1}^{n} \left\| \boldsymbol{x}^{(i)} - \boldsymbol{\mu}_x \left( \boldsymbol{z}^{(i)}; \theta \right) \right\|_2^2 \right) + \frac{1}{d} \sum_{k=1}^{\kappa} h \left( \frac{1}{n} \| \boldsymbol{z}_k \|_2^2 \right), \qquad (22)$$

where in the present context $\kappa = d = 2 > r = 1$, and as mentioned previously, $\boldsymbol{z}_k$ represents the $k$-th row of $\boldsymbol{Z}$. This arrangement is equivalent to simply assuming that the encoder is capable of computing the minimizing $\boldsymbol{z}^{(i)}$ for each index (i.e., we have removed amortized inference). We adopt this assumption for simplicity of exposition, but the same conclusions can be drawn in broader conditions.

To achieve zero reconstruction under the stated conditions using only $r = 1$ nonzero row of $\boldsymbol{Z}$, we must choose $\theta = 1$. In this restricted setting, the optimal $\boldsymbol{Z}$ will satisfy $\frac{1}{n}\|\boldsymbol{z}_1\|_2^2 \geq \frac{1}{n}(nC^2) = C^2$ and $\frac{1}{n}\|\boldsymbol{z}_2\|_2^2 = 0$ such that the overall objective value will be

$$\min_{\phi} \mathcal{L}_{g,h}(\theta = 1, \phi \in \Phi) \;=\; g(0) + \frac{1}{2}\left[h(0) + h\left(\tfrac{1}{n}\|\boldsymbol{z}_1\|_2^2\right)\right] \;\geq\; g(0) + \frac{1}{2}h(0) + \frac{1}{2}h\left(C^2\right), \quad (23)$$

where $\Phi$ is the set of $\phi$ values that lead to zero reconstruction error. In other words, within the current setup, the constraints $\theta = 1$ and $\phi \in \Phi$ are necessary conditions for any solution to achieve an optimal sparse representation, and such a solution will incur a cost of at least $g(0) + \frac{1}{2}h(0) + \frac{1}{2}h\left(C^2\right)$.

But now suppose we choose $\theta = 0$. In this revised situation, the optimal unconstrained $\boldsymbol{Z}$ will have rows satisfying $\frac{1}{n}\|\boldsymbol{z}_k\|_2^2 \leq \epsilon^2$, $k = 1, 2$. The associated cost then becomes

$$\min_{\phi} \mathcal{L}_{g,h}(\theta = 0, \phi) \;=\; g(0) + \tfrac{1}{2}\sum_{k=1}^{2} h\left(\tfrac{1}{n}\|\boldsymbol{z}_k\|_2^2\right) \;\leq\; g(0) + h\left(\epsilon^2\right). \quad (24)$$

At this point, without loss of generality assume that $h\left(C^2\right) = 1$ and $g\left(0\right) = h\left(0\right) = 0$, which can be accomplished by simply translating and rescaling the overall cost. Because $\lim_{u \to 0^+} h'(u)$ is bounded, the gap between $h(0)$ and $h\left(\epsilon^2\right)$ can be made arbitrarily small for $\epsilon$ sufficiently small. In contrast, the gap between $h\left(\epsilon^2\right)$ and $h\left(C^2\right)$ can be arbitrarily close to one. Therefore, it follows that if our data was generated with $\epsilon$ sufficiently small, then

$$\min_{\phi} \mathcal{L}_{g,h}(\theta = 1, \phi \in \Phi) \;\geq\; \tfrac{1}{2} \;>\; \min_{\phi} \mathcal{L}_{g,h}(\theta = 0, \phi) \;\approx\; 0, \quad (25)$$

and so the unique solution achieving zero construction error with a single active latent variable cannot be the global optimum. Or equivalently, any globally optimum solution will not coincide with an optimal sparse representation. And finally, once the basic result has been established for the lowest possible dimensions satisfying $d \geq \kappa > r > 0$, it is straightforward to extend to higher dimensions by concatenating replicas of the 2D case and/or incorporating additional linear projections as needed (for both data generation and decoding).

Note that the situation would be completely different if $h(u) = \mathcal{I}[u > 0]$, meaning an indicator function that equals zero if $u = 0$ and one for all $u > 0$. In this case, it is clear that $\min_{\phi} \mathcal{L}_{g,h}(\theta = 1, \phi \in \Phi) = \frac{1}{2}$ while all other solutions with zero reconstruction error will be such that $\mathcal{L}_{g,h}(\theta, \phi) = 1$. But of course this $h$ does not have a bounded gradient everywhere because of the discontinuity at zero.

Additionally, while the proof is obviously predicated on a counterexample designed with a specific technical purpose in mind, it is nonetheless emblematic of situations that may naturally arise in practice. For example, it is easy to envision scenarios where data is lying on a complicated $r$-dimensional manifold that is contained within a larger $(r + s)$-dimensional manifold (or possibly subspace) that has much simpler structure (here we are assuming $r < r + s < d$). Perfectly reconstructing such data could be accomplished using either (a) only $r$ dimensions, or (b) using $(r + s)$ dimensions, depending on whether the complex or simple manifold was modeled by the decoder. But unless we have a penalty function with a strong preference for lower-dimensional structures, then the network may well favor or converge to the simpler, higher-dimensional alternative. ∎