# OpenReview forum: "On the Value of Infinite Gradients in Variational Autoencoder Models"
_NeurIPS.cc/2021/Conference — NeurIPS 2021 Spotlight_

### Official Review · Reviewer_VPWz · 2021-06-28

**Rating:** 6
**Confidence:** 3

**Summary:**

This paper studies the infinite gradient of training VAEs, namely when the decoder's variance is a learnable parameter, the variance will converge to zero, resulting in unbounded gradient. Although previously this is seen as an undesirable property, the authors prove that unbounded gradients are necessary for obtaining optimal spare representations. The authors also try to argue that learning the variance is better than fixing it to an arbitrarily small value.

**Limitations And Societal Impact:**

Yes.

**Main Review:**

The paper address an important issue in VAE training. It is widely observe that when the decoder variance is learnable, it will converge to a small number near zero. This paper make a nice connection between this phenomenon and optimal sparse representations, which is arguably the goal for learning an AE or VAE. By proving that it is inevitable to avoid the unbounded gradient while minimizing a generalization of AE loss, the adaptive decoder variance is justified.

I have several questions regarding the paper.

1. When you study the learning of decoder variance, do you consider the effect of initialization? Typically when we learn the decoder variance, it is initialized as 1. However, what if we use other initializations, for example, initialize it at the convergence value, but still set it to be learnable? Maybe it is interesting to study different initializations using the dynamics in figure 4b.

2. I have a question regarding your statement that the using learnable gamma is important for improving sample quality. However, in my understanding, since learnable gamma will result in very small value of gamma, in the end the VAE model is still somewhat under regularized, and directly sample from it should not lead to good sample quality. Previously this is resolved by training another generative model on the aggregated posterior, but I think the latent model is something else, and strictly speaking the sample obtained by two-stage models are not samples from the VAE. Maybe the authors can discuss a bit on this.

3. Regarding the OOD detection experiments, since the log likelihood (or equivalently BPD) of the VAE is largely dominanted by the reconstruction loss, and you show that learning gamma and fixing gamma at a small value leads to similar recon loss, do you observe similar performance of OOD detection when gamma is fixed at a small value? Intuitively they should behave similar here.


After rebuttal:
Thanks for addressing my concern, I will keep my score.


**Time Spent Reviewing:**

2

---

> ### Author Response · Authors · 2021-08-10
> **Response to Reviewer VPWz**
>
> Thanks for the constructive comments and recognizing that our work addresses an important aspect of VAE models.  We address reviewer questions as follows.
>
> **Question:**  When you study the learning of decoder variance, do you consider the effect of initialization? Typically when we learn the decoder variance, it is initialized as 1. However, what if we use other initializations, for example, initialize it at the convergence value, but still set it to be learnable?
>
> **Response:**  If the decoder variance is initialized to a small value, then it will generally just quickly move up to a larger value at the beginning of the optimization trajectory.  This is because before the other decoder and encoder parameters have converged, the reconstruction error will be large, and therefore the optimal $\gamma$ as given by Eq. (5) will also be large.
>
> **Question:**  I have a question regarding your statement that the using learnable $\gamma$ is important for improving sample quality. However, in my understanding, since learnable $\gamma$ will result in very small value of $\gamma$, in the end the VAE model is still somewhat under regularized, and directly sample from it should not lead to good sample quality. Previously this is resolved by training another generative model on the aggregated posterior, but I think the latent model is something else, and strictly speaking the sample obtained by two-stage models are not samples from the VAE. Maybe the authors can discuss a bit on this.
>
> **Response:**  It is true that for data lying on a manifold, the VAE loss can be dominated by the reconstruction loss as $\gamma$ becomes small, and therefore as the reviewer suggests, a second generative model trained on the aggretated posterior can be helpful.  But while this second stage model may improve the quality of generated samples (when using ancestral sampling applied to both models), the role of finding optimal sparse representations remains with the original first stage, which in turn requires a learnable $\gamma$.
>
> **Question:**  Regarding the OOD detection experiments, since the log likelihood (or equivalently BPD) of the VAE is largely dominanted by the reconstruction loss, and you show that learning gamma and fixing gamma at a small value leads to similar recon loss, do you observe similar performance of OOD detection when gamma is fixed at a small value? Intuitively they should behave similar here.
>
> **Response:**  The performance may be similar assuming the models somehow converge to the same final solution; however, similar reconstruction error alone does not imply that the final solutions are wholly similar, e.g., as evidenced by the discussion in Section 4 related to Figure 4(a).  More specifically, even if the reconstruction errors are similar, there is still value in forcing the latent space to use the fewest number of information dimensions to achieve a given reconstruction error.  For example, as shown in the Figure 1 toy example, PCA can achieve zero reconstruction error, and yet it is more likely to accurately reconstruct outliers as well because it does not produce the optimal sparse representation that exists within the curved 1D manifold.

---

### Official Review · Reviewer_ATkA · 2021-07-16

**Rating:** 6
**Confidence:** 3

**Summary:**

**Update after the authors' response**: I appreciate the detailed authors' response, and while it clarifies some points the paper remains a borderline case to me unless the applicability of the findings in the paper was clearly acknowledged and the breadth of the main claims was reduced accordingly.

**Update after further engagement with authors**: We have managed to clear up some misunderstanding / misinterpretation from both sides, and I have raised my score accordingly from a 5 to a 6.

The paper investigates Autoencoders with a Gaussian likelihood model, where the variance is a trainable parameter. As has been observed before, this often leads to learned variances approaching zero (particularly for high-capacity models and noise-free data), which in turn leads to exploding gradients. The paper argues that these gradients and the collapse of the variance towards very low values should be embraced, rather than being seen as a pathology of the model. In particular the paper argues that under Gaussian likelihood and a fixed latent-space prior, if the goal is to learn a lossless representation of the dataset (which implies a high-fidelity generative model) near unbounded gradients are necessary to avoid underfitting (i.e. using unnecessary latent dimensions). Finally the paper proposes a scheme to mitigate optimization issues with unbounded gradients and illustrates the scheme on CelebA

**Contributions:**

1) Analyzing the origin of unbounded gradients of Gaussian likelihood (V)AEs with learnable (co-)variance. Significance: unsurprisingly, the source of the problem is the Gaussian likelihood term, which is unbounded for the variance approaching zero. The latter has been pointed out various times in the VAE literature and is a well known pathology when fitting Gaussian (mixture) models.

2) A definition of optimal sparse representations as allowing for zero reconstruction error while only using a minimally required amount of dimensions (lossless compression). Significance: I think the definition can be useful for some particular applications (most notably training a high fidelity generative model), but in practice I would argue there are many clear cases where zero reconstruction error is clearly undesirable for an optimal representation (because the data contains noise or other task-irrelevant variations that should not be captured by the representation). This is well discussed in the literature on lossy compression and representation learning (rate distortion theory and information bottleneck literature and their modern application to neural networks and auto-encoder-like architectures).

3) Showing that, under said definition, “optimal sparse representations” are only attainable via near-unbounded gradients for any AE type architecture. Significance: I think this claim is at best overstated, at worst wrong: since Gaussian likelihoods are the root of the gradient explosion, the findings do not apply for other likelihood models that do not suffer from the pathology of unboundedenss (under a certain parametrization), such as e.g. Bernoulli likelihoods. The claim thus only applies to Gaussian likelihoods and other likelihood models with unbounded values, but not any AE architecture. Please correct me if my understanding (on the reliability of the findings on Gaussian likelihoods) is wrong.

**Ethical Concerns:**

No immediate ethical concerns. It would be nice to add a short statement regarding ongoing ethical debates with high-fidelity generative models, to which this work contributes (mostly through analysis though).

**Limitations And Societal Impact:**

Please add a paragraph with a focused discussion of the limitations of the proposed method (and perhaps reiterating the limited scope of the claims / theoretical analysis).

**Main Review:**

**Originality, Quality, Clarity, Correctness:**

The paper follows up on a previously pointed out pathology, and shows that this pathology is unavoidable and rather than working around it (by limiting gradient magnitude), it should be incorporated into the optimization process (leading to a variance-smoothing scheme). The technical part of the paper is well written, but important conceptual assumptions are not discussed (many works would disagree that optimal representations are lossless), and some claims are made w.r.t. arbitrary AE models where, as far as I can tell, all the claims strictly only apply to Guassian-likelihood models (and other likelihoods that grow unbounded when fitting noise-free data) with fixed latent-space prior distribution (i.e. no aggregate posterior). Some of the claims (please correct me if I’m wrong) no longer apply when either of these assumptions is not met. The experiments shown are insightful, but since only a single experiment with the proposed variance-smoothing scheme is shown, it remains somewhat unclear how well the  corresponding results generalize and lead to claimed improvements - particularly since comparisons to other methods are missing and the sparsity of the solution obtained is not computed.

**Verdict:**

Overall I am on the fence with this paper. The analysis shown in the paper seems correct, but the findings are not very surprising, given that the likelihood model is Gaussian. Some of the claims (no optimal representation with any AE without near-unbounded gradients) sound very surprising, but only because they are stated without the explicit assumptions under which they hold. My two main concerns (see more details below) are: one, the analysis and results seem not to apply to well behaved likelihood models - this is never stated in the paper and should be clearly spelled out. Two, the definition of optimal representations (as lossless compressions) is too broad - it might be appropriate in some tasks, but it is clearly inappropriate in other tasks (where lossy compressions are desirable, e.g. for generalization, or under noisy data). This needs to be discussed thoroughly. I am therefore leaning towards rejection, noting that these issues can potentially be sufficiently addressed in the rebuttal. However since these main concerns regard multiple passages throughout the paper it is hard to judge whether a revised version of the manuscript would address the criticism sufficiently, without seeing the actual revised version. Either way, I am happy to revise my final verdict based on the other reviews and the authors’ response.

**Pros:**
 * Thorough analysis of the main problem and pointing out that under a number of assumptions the main problem cannot be avoided
 * Proposal on how to incorporate the problem into a more numerically stable optimization process.
 * Concrete definition of the goal (optimal sparse lossless representations)

**Cons:**
 * Claims stated in very general terms, despite depending on fairly specific assumptions (which, admittedly, are standard modelling assumptions in large parts of the VAE literature).
 * Main pathology results from unbounded likelihoods which has been identified as problematic before. Discussion of these previous works falls a bit short.
 * Experimental findings for the proposed variance-smoothing look OK but there is not a lot of experimental evidence to back them up, and the single experiment that was run does not have multiple repetitions such that the stat. significance of the empirical findings remains unclear (though I do expect most findings to be significant, the margins look fairly good).

**Improvements:**

1) I disagree with Definition 1 (in the broadest sense): optimal sparse representations for many tasks are produced via **lossy** compression, implying non-zero construction error. This includes tasks like representation learning, disentanglement, outlier detection with noisy data - in fact any task that assumes that the data is not noise-free (hence each datapoint contains some irrelevant information which should be removed via lossy compression). There are numerous papers the topic of what constitutes good representations, perhaps most notably
 * The Information bottleneck method, Tishby et al. 2000
But also:
 * Emergence of Invariance and Disentanglement in Deep Representations, Achille et al. 2018
 * Fixing a broken ELBO, Alemi et al. 2018.
 * Deep Variational Info Bottleneck, Alemi et al. 2017
This needs to be discussed in at least a paragraph, and the definition needs to be weakened to “optimal lossless representations for noise-free data” or similar (I’m not one of the authors, so certainly not asking for citations but would like to see a proper discussion).

1a) Clearly point out in which applications the lossless compression is desirable, and in which applications it is not desirable. Section 2.2 is currently biased towards one side.

1b) Outlier detection with low \gamma (and the illustration in Fig. 1) assumes (almost completely) noise-free data that lies on a smooth manifold perfectly captured by the model (i.e. perfect inter- and extrapolation). This rarely holds in practice and should be discussed in a more nuanced fashion.

2) Please (throughout the paper) make it clear that the main claims only apply to models with Gaussian likelihood (and trainable variance, but the latter is made sufficiently clear in the paper). Some of the claims sound like they would apply to any AE with arbitrary likelihood models. This is not meant to reduce the significance of the paper (after all I would say that the paper addresses the most widely used VAE model, hence it is of high significance) but to sharpen the focus and generalizability of the claims and results. I personally also think that the community should think more carefully about the standard modeling assumptions and search for alternative likelihood models whenever appropriate. The latter message is missing completely from the paper.

3) If the goal is to learn a high-fidelity image generator (or other generative model) or density modeling of noise-free data then a reconstruction error of 0 is indeed desirable, however I personally would question whether variational autoencoders are the right choice in these situations (admittedly they have been used in the literature in this context). It should be fairly obvious that a Gaussian likelihood with learnable variance is a wrong modelling choice in such a situation (I would even argue that having a fixed prior is often a wrong modelling choice, particularly when underfitting is an issue). Alternatives should be discussed (e.g. flow-based models, Bernoulli likelihoods, “aggregate posteriors” as priors, …). Please make an effort here to point out alternatives, I am convinced that readers will greatly benefit from that and it will help put the paper into wider context.

4) Sparsity as discussed in Definition 1 (ii) (line 105) is achieved by pushing the posterior towards the prior for non-informative dimensions. It has been discussed before (info bottleneck literature, fixing a broken ELBO) how this can be achieved by implementing a proper lossy compression scheme (following rate distortion theory) where, crucially, the prior is not fixed but given by the marginal or “aggregate posterior”. Famously, this leads to unneeded dimensions not being used (their distribution is the same as the prior which is the aggregate posterior). I therefore disagree with Line 327: “Hence there is no obvious alternative if this form of parsimony is our goal” - sparsity can also be achieved by using an aggregate posterior. This is also what Tomzcak and Welling 2018 do (but also e.g. Deep Variational Info Bottleneck which uses a variational approximation to the marginal). Please discuss/clarify.


**Comments:**

 * Line 345: I only partially agree. Adaptable/trainable priors should help with getting sparse representations - if they use Gaussian likelihoods they will not prevent infinite gradients of course (see also Improvement 4) above).
 * Line 219-220: Given my comments above (in the improvements) I think this statement needs to weakened or it needs to be clarified under what assumptions the statement holds (Gaussian likelihoods, fixed piors, and the precise definition of sparse optimal representation as Definition 1, but not for other, e.g. lossy optimal representations).


**Time Spent Reviewing:**

4.5

---

> ### Author Response · Authors · 2021-08-10
> **Response to Reviewer ATkA**
>
> We sincerely appreciate the reviewer's very detailed, comprehensive comments. However, because the review is quite lengthy and raises related points in various places (e.g., summary, main review, improvement recommendations sections, etc.), we have chosen to just address the primary concerns, grouping our response by comment category rather than directly point-by-point in sequential order.  Hopefully this will best facilitate further discussion if needed while maintaining brevity.
>
> **Questions Regarding the Definition of Optimal Sparse Representations**
>
> In Definition 1 we formally define the particular notion of optimal sparse representation that is applied throughout our work.  Of course we completely agree with the reviewer that other reasonable definitions can be proposed as well depending on the application scenario being considered.  However, we chose our stated version in part for analysis purposes, and in part because for many types of data lying on or at least very near low-dimensional manifolds (which is essentially all natural images), our definition of an optimal sparse representation directly aligns with the dimensionality of such manifolds.  And Definition 1 also corresponds with a long precedent in the signal and image processing literature involving the search for low-dimensional structure in high-dimensional data.  We can include additional supporting references to a revision, e.g., seminal papers such as "Optimally sparse representation in general (nonorthogonal) dictionaries via $\ell_1$ minimization," "Robust uncertainty principles: Exact signal reconstruction from highly incomplete frequency information," and "Exact matrix completion via convex optimization" among many others.
>
> Moreover, as mentioned in Footnote 1 of our submission, we can in principle relax the requirement of perfect reconstructions to accommodate some degree of approximation error.  In this scenario, we envision that the basic intuitions elucidated in our paper would still apply, even if the exact assumptions of our main technical results do not strictly hold. This is common practice in analysis, whereby simplified noise-free conditions are assumed for tractability.
>
> Regardless, we can address alternative notions of optimality in learning representations (as well as additional citations as needed, thanks for the pointers) in a revision as space allows, which might help to alleviate reviewer concerns.
>
> **Lossy Versus Lossless Representations**
>
> Just to clarify further, in using Definition 1 our goal is not to promote lossless representations as somehow more relevant to the valid lossy alternatives mentioned by the reviewer.  Rather, our point is that many data types such as natural images have well-established low-dimensional structure that can be closely approximated by lossless sparse representations, and therefore, analysis of such representation (even if admittedly idealized) nonetheless provides useful insights into the behavior of VAEs and broader related models.  Note also that the observation in multiple prior works (as cited in our paper) that the learned variances from the Gaussian VAE decoder can converge towards zero directly supports the notion that near perfect reconstructions are possible and that the underlying data is indeed very near a low-dimensional manifold.  Please see Eq. (5) in our submission which shows the optimal value of the learned variance and its relationship to reconstruction error.
>
> **Gaussian Versus Other Likelihood Models**
>
> As we have stated throughout the submission, our work is devoted to VAEs applied to continuous data, for which Gaussian likelihood models are by far the most common assumption.  We are not aware of other continuous data likelihoods being regularly applied to VAEs in practice, but even so, the core conclusions we arrive at would likely still hold, noting that the AE model we examine is not strictly based on a Gaussian likelihood anyway (or the negative logorithm thereof; more on this below).  Moreover, for data lying on or near a low-dimensional manifold, any possible likelihood model that correctly restricts its probability mass to the manifold will necessarily involve an unbounded density and thus enter the scope of our analysis.
>
> Additionally, while the reviewer mentioned a Bernoulli likelihood as a candidate alternative, which has at times been heuristically applied to image data in some papers, this is not actually a valid distribution for continuous data.  But even if we accept the lack of a valid underpinning probabilistic model, it is not difficult to show that optimal sparse representations are still  not generally possible using the Bernoulli unless an unbounded scaling is applied that merely reintroduces unbounded gradients.  Consequently, given all of the above, we do not believe the Gaussian likelihood assumption is especially problematic, at least within the stated scope of continuous VAE models.
>
> **Fixed Versus Trainable Priors**
>
> While in practice trainable priors can be effective, when we revert back to the original latent variable model presumed to underlie the data, we can generally (at least in principle) convert  a model with a trainable prior to an equivalent one with a fixed prior.  For example, if the original model with trainable prior is $p_\theta(x) = \int p_\theta(x|z)p_\theta(z) d z$, then we can express the parameterized prior as $p_\theta(z) = \int p_\theta(z|u)p_\theta(u) d u$ with additional latent variable $u$. This naturally leads to a reparameterized version of $p_\theta(x)$ based on a fixed, parameter-free prior via
> $$p_\theta(x) = \int p_\theta(x|z)p_\theta(z) d z = \int\int p_\theta(x|z)p_\theta(z|u) p(u) d z du = \int p_\theta(x|u) p(u) du,$$
> where now $p_\theta(x|u)$ can be viewed as simply a more complex/flexible decoder than the original $p_\theta(x|z)$ (with some abuse of notation we use $\theta$ for the parameters in all of these models).  So in this sense, we believe that our results apply equally well to models with flexible priors, as the issue of infinite gradients being needed to obtain optimal sparse representations will still hold as before.  That being said, trainable priors can still of course be useful for matching the aggregate posterior to the prior and generating good samples, but this is a separate issue from obtaining optimal sparse representations, and is out of the scope of this work.
>
> **Some of the claims sound like they would apply to any AE with arbitrary likelihood models**
>
> Our Theorem 4 is expressed with respect to the particular form of AE model described by Eq. (6).  However, this form was largely chosen because it facilitates a direct bridge with the VAE.  In fact, Theorem 4 still holds when the function $h$ applied to the data fit term in Eq. (6) is changed to some arbitrary alternative function $g$ (i.e., it is no longer shared with the regularization term).  Or analogously, it holds for any negative log-likelihood function that is sufficient for enforcing zero reconstruction error in the sense of $x^{(i)} = \mu_x(z^{(i)};\theta)$ for each training sample.  Inspection of the proof reveals that enforcing a shared function $h$ on both AE terms is not actually needed.  Hence our results do essentially apply to any AE model capable of producing perfect reconstructions.  We can clarify these points in a revision; thanks for pointing out this issue.
>
> **Other More Minor Question**
>
> While we cannot include additional figures here, the results shown in Figure 4 are complementary to other results and illustrations in our main paper and supplementary, and are also stable across different testing regimes we have tried.

---

> > ### Comment · Reviewer_ATkA · 2021-08-18
> > **Thanks for the detailed response**
> >
> > I want to thank the authors for their detailed response.
> >
> > I acknowledge and understand the authors' responses - and I agree with most of the response. However, some of my main criticism remains (which is not about the particular technical findings but about the main narrative and the breadth of applicability of the claims). I do think that the paper at its core is technically correct, but the verbal claims made throughout the paper need to be more precise and limited to the setting studied. I will try to clarify as best as I can below.
> >
> > First, I think there is perhaps a bit of a clash in framing. The authors seem to mainly be inspired by (1) **noise-free** image-data (which lies on a low-dimensional manifold) where (2) the **goal of representation learning is to capture this manifold** by (3) **minimizing a (log) likelihood** using (4) **gradients**. This is very useful for e.g. using the learned representation in a generative model that can produce high-fidelity images that are not in the training set. However, such a representation is less useful e.g. for training a downstream MNIST classifier (the information content of an optimal representation for classification is simply digit-identity, which would imply large pixel-level reconstruction errors). Similarly, if the data is considered to be noisy (e.g. any real-world image data has camera noise, which in many applications has become negligible recently) optimal representation should **not** capture this noise (thus leading to improved generalization). Famously, one of the earliest techniques to train auto-encoders was to actually inject noise (denoising AEs) into the input, to prevent the AE from learning a trivial representation (that allows for zero-reconstruction error).
> >
> > I would be happy with the paper if the setting ( (1)-(4) ) were clearly discussed (and alternatives mentioned throughout) and verbal claims were clearly limited to this setting only - but the current version of the manuscript makes very broad claims, which do not hold under noisy data, error functions other than a log-likelihood with a trainable precision parameter, and goals/downstream tasks that do not require lossless representations. While the authors' response somewhat acknowledges that these regimes exist, there seems to be little willingness to change the manuscript and breadth of the claims.
> >
> >
> > > Hence our results do essentially apply to any AE model capable of producing perfect reconstructions.
> >
> > Again, this claim assumes that training is done via minimizing a log-likelihood with trainable precision. If data can be decomposed into some basis (e.g. binary images resulting from the additive combination of some base-functions) then e.g. a plain old "sparse overcomplete" AE that uses this basis can achieve near-zero reconstruction error. I understand that strictly speaking this does not fall under the models considered in the paper (since there is no trainable precision parameter) - but that only becomes clear from looking at the technical details, the verbal claims alone are too broad.
> >
> > > Fixed vs Trainable Priors
> >
> > I agree that trainable prior-models can be translated into fixed-prior models, but this misses the point of my comment. The "aggregate posterior" is a principled way of ensuring that unnecessary dimensions of the representation are unused (i.e. they are pushed towards the prior). Thus, I disagree with Line 327: “Hence there is no obvious alternative if this form of parsimony is our goal”. This is another example of an overstated claim - there are alternatives which should be mentioned in the paper, and the paper could clearly discuss pros and cons of the proposed approach against these alternatives. That's what I would like to see done thoroughly throughout the paper.
> >
> > > We are not aware of other continuous data likelihoods being regularly applied to VAEs in practice
> >
> > What about flow-based models? What about mixture-of-Gaussian likelihoods (which admittedly suffer from similar problems) and other hierarchical models? I somewhat agree with the statement above when looking at image generation as the main application, but VAEs are also used for many other tasks.

---

> > > ### Author Response · Authors · 2021-08-23
> > > **Follow-Up to Reviewer ATkA**
> > >
> > > **Comment:**
> > > I acknowledge and understand the authors' responses - and I agree with most of the response. However, some of my main criticism remains (which is not about the particular technical findings but about the main narrative and the breadth of applicability of the claims). I do think that the paper at its core is technically correct, but the verbal claims made throughout the paper need to be more precise and limited to the setting studied.
> > >
> > > **Response:**
> > > We appreciate the reviewer taking the time to go through our detailed author rebuttal and continuing to engage with this paper.  And as the reviewer has acknowledged, the remaining point of contention is not about technical correctness of stated results per se, but rather, the breadth of applicability and framing of these results.  We will address this issue point-by-point below (abbreviating some reviewer comments to fit within the space limits).
> > >
> > > **Comment:**
> > > First, I think there is perhaps a bit of a clash in framing. The authors seem to mainly be inspired by (1) noise-free image-data (which lies on a low-dimensional manifold) where (2) the goal of representation learning is to capture this manifold by (3) minimizing a (log) likelihood using (4) gradients. This is very useful for e.g. using the learned representation in a generative model that can produce high-fidelity images that are not in the training set. However, such a representation is less useful e.g. for training a downstream MNIST classifier...
> > > Similarly, if the data is considered to be noisy ...
> > >
> > > **Response:**
> > > While the four points the reviewer mentioned do align with the content of our paper, we might still frame the narrative slightly differently. As described in the abstract and again in Section 1, the starting point for this work is the frequent observation in the literature that when training VAE models on continuous data, the gradients sometimes become large, possibly even unbounded.  And as we further discuss, and prior work has noted as well, this occurs because for data lying on a low-dimensional manifold (or at least nearly so), not only will the true data log-likelihood be unbounded, but the VAE ELBO as well when granted sufficient capacity for encoder and decoder modules. (And this is true even with Gaussian encoder and decoder assumptions, provided the mean networks are sufficiently complex.)
> > >
> > > Within this broader context, the remainder of our submission is then largely devoted to the narrower task of explaining: (i) How AE-based loss functions, with unbounded gradients around minimizing solutions like the VAE, can serve a useful role in obtaining optimal sparse (lossless) representations per Definition 1, and (ii) how a learnable decoder variance parameter can potentially help to mitigate the impact of such gradients during training.  Of course we completely agree with the reviewer that the particular type of sparse representation we analyze is not relevant for many downstream tasks, and that there are numerous scenarios where different notions of optimality would be more appropriate. But these scenarios don't necessarily follow from our starting point, which is the observation of unbounded gradients and continuous data on low-dimensional manifolds.
> > >
> > > **Comment:**
> > > I would be happy with the paper if the setting ( (1)-(4) ) were clearly discussed (and alternatives mentioned throughout) and verbal claims were clearly limited to this setting only - but the current version of the manuscript makes very broad claims, which do not hold under noisy data, error functions other than a log-likelihood with a trainable precision parameter, and goals/downstream tasks that do not require lossless representations. While the authors' response somewhat acknowledges that these regimes exist, there seems to be little willingness to change the manuscript and breadth of the claims.
> > >
> > > **Response:**
> > > Perhaps there was some miscommunication on our part in the initial author rebuttal, but we are more than happy to adjust the manuscript in any areas where the breadth of claims should be adjusted. Indeed, it serves us no good purpose to make claims that are not supported by the actual technical content of the paper.  Additionally, the more detailed responses below may also help in addressing the specific individual points.
> > >
> > > And as mentioned in Sections 1 and 2, our focus is on continuous data lying on a low-dimensional manifold, and the value of infinite gradients in obtaining lossless representations on such manifolds.  That alternative lossy representations are well-motivated and serve many useful purposes is of course true, just not the focus of this paper.  Perhaps one mistake we made was not explicitly stating this fact up front.
> > >
> > > **Comment:**
> > > "Hence our results do essentially apply to any AE model capable of producing perfect reconstructions."  Again, this claim assumes that training is done via minimizing a log-likelihood with trainable precision. If data can be decomposed into some basis (e.g. binary images resulting from the additive combination of some base-functions) then e.g. a plain old "sparse overcomplete" AE that uses this basis can achieve near-zero reconstruction error. I understand that strictly speaking this does not fall under the models considered in the paper (since there is no trainable precision parameter) - but that only becomes clear from looking at the technical details, the verbal claims alone are too broad.
> > >
> > > **Response:**
> > > Just to clarify, our results do actually apply to AE models that are *not* constructed as a log-likelihood with a trainable precision.  For example, please note that Theorem 4 relates to AE models with arbitrary penalties $h$, which need not be related to any log-likelihood with a trainable precision (indeed, $h$ has no trainable parameters at all).  And as we stated in our initial author rebuttal, Theorem 4 still holds when the function $h$ applied to the data fit term in Eq. (6) is changed to some alternative function $g$ (i.e., it is no longer shared with the regularization term).  It is also worth mentioning that a sparse overcomplete AE that achieves zero reconstruction error has not necessarily achieved an *optimal* sparse representation per our Definition 1.  For this, it must also use the fewest possible number of nonzero basis coefficients.
> > >
> > >
> > > **Comment:**
> > > "Fixed vs Trainable Priors."  I agree that trainable prior-models can be translated into fixed-prior models, but this misses the point of my comment. The "aggregate posterior" is a principled way of ensuring that unnecessary dimensions of the representation are unused (i.e. they are pushed towards the prior). Thus, I disagree with Line 327: “Hence there is no obvious alternative if this form of parsimony is our goal”. This is another example of an overstated claim - there are alternatives which should be mentioned in the paper, and the paper could clearly discuss pros and cons of the proposed approach against these alternatives. That's what I would like to see done thoroughly throughout the paper.
> > >
> > > **Response:**
> > > Perhaps we misunderstood the reviewer's point here, but we do not see any clear way that trainable priors somehow allow a model to produce an optimal sparse representation (per our definition) while avoiding infinite gradients.  Indeed a VAE model, even with a trainable prior, will still generally have infinite gradients around optimal sparse lossless representations of continuous data lying on a low-dimensional manifold. (And note that unnecessary dimensions converging to the prior is not actually the source of infinite gradients; rather it is the needed dimensions converging to near zero variance and the corresponding near-zero reconstruction errors that ensue.)  Additionally, our original statement the reviewer references from line 327, “Hence there is no obvious alternative if this form of parsimony is our goal," was simply meant to imply that there is no obvious alternative that avoids such unbounded gradients.
> > >
> > > From another viewpoint, consider the parameterized prior $p_\theta(z)$ defined such that $z = f_\theta(u)$ and $u\sim N(0,I)$, where $f_\theta$ an arbitrary invertible and differentiable function.  Then the corresponding VAE ELBO can be converted to an equivalent VAE ELBO with fixed prior $N(0,I)$ and updated posterior $\tilde{q}_{\phi,\theta}(z|x)$.  The latter is defined via $z = f_\theta^{-1}(u)$ with $u\sim q_\phi(u|x)$, where $q_\phi(\cdot |x)$ is the original posterior. (Note that the KL term is invariant to such a transformation applied to both prior and posterior, and the VAE reconstruction term can absorb the transformation $f_\theta$ into the decoder). Hence the profile of infinite gradients is the same as an equivalent VAE model with no trainable prior.  And in either case there will be infinite gradients around solutions that produce optimal sparse lossless representations.  But again, we may have misinterpreted the reviewer's comment here, so please let us know if further clarification is warranted.
> > >
> > >
> > > **Comment:**
> > > "We are not aware of other continuous data likelihoods being regularly applied to VAEs in practice."  What about flow-based models? What about mixture-of-Gaussian likelihoods (which admittedly suffer from similar problems) and other hierarchical models? ...
> > >
> > > **Response:**
> > > Fair point.  We had previously thought that flow-based models were more commonly applied to the encoder to create a more accurate posterior approximation.  But if they are also used for defining a richer class of decoder likelihoods, then this is worth noting (any specific references for this that the reviewer suggests?).  That being said, if the data truly lies on a low-dimensional manifold, then a richer decoder likelihood model (flow-based or otherwise) will only increase the chances that an unbounded likelihood with infinite gradients is achievable, hence the basic notions in our submission are still relevant.

---

> > > > ### Comment · Reviewer_ATkA · 2021-08-23
> > > > **Thanks for the continued engagement**
> > > >
> > > > I want to thank the authors for their engagement and detailed response. I think that after two rounds of engagement we have managed to clarify our differences and misunderstanding. I do think that other readers with a background in Bayesian DL, and (variational) representation learning might have a similar reading to my initial reading (note that there was some overlap in my criticism and other reviewers' criticism).
> > > >
> > > > I am on board with the main narrative if it is clearly expressed in paper and the alternatives (even if they are outside the scope of the paper) are discussed well. I think that would essentially sound similar to the authors' response above by adding some caveats (highlighting the limitations in focus early on in the paper) and sharpening a few passages. I am still no terribly happy with claims such as: "finding X applies to any (V)AE model" - what's much better is that "under our definition of an optimal sparse lossless representation, any continuous (V)AE model with large enough capacity suffers from large/unbounded gradients if the data lies near a low-dimensional manifold (near noise-free data)". And then a small paragraph in the discussion that says that under other definitions of optimal lossy representations (which are relevant for e.g applications X, Y, Z), unbounded gradients can be avoided. I do think that the authors have understood this criticism after our engagement, and want to apologize for not expressing this clearly enough initially.
> > > >
> > > > Re: trainable priors - I agree that trainable priors **under said definition of optimal lossless sparse representations** do not avoid unbounded gradients. To me trainable priors are one (principled) way of producing sparse (lossy) representations with VAEs - hence my disagreement (because I read the claim in isolation relating to sparsity only, and not conditional on the definition introduced earlier). Thanks for clearing this up.
> > > >
> > > > I think the initial manuscript needs some more polishing (in terms of presentation and discussion, not necessarily on the technical level), but I also think that the authors are in a good position to do this polishing. I will raise my score accordingly and will not oppose accepting the paper - unfortunately since it is hard to judge the end-result without seeing the revised manuscript (which is unfortunately not possible at NeurIPS) I cannot give a very high score with high confidence.

---

> > > > > ### Author Response · Authors · 2021-08-25
> > > > > **Response to Reviewer ATkA**
> > > > >
> > > > > This exchange was quite useful for refining the message of the paper, thanks again for all the detailed feedback and time invested in this review.  And we will definitely take these discussion points into account in preparing an improved revision.

---

### Official Review · Reviewer_69AE · 2021-07-17

**Rating:** 5
**Confidence:** 5

**Summary:**

Summary. This paper tries to analyze the regularization trade-offs for variational auto-encoding (VAE) models, in particular, explaining the infinite gradient phenomena in the training of VAE models. The author(s) defines some concepts to carry out the analyses.

**Limitations And Societal Impact:**

- Infinite gradients or overfitting? It turned out the infinite gradient issue the author(s) heavily focused on is caused by letting the precision parameter of the Gaussian noise model be trainable, which always explodes to infinity. Well, it is well known this is simply the overfitting in the finite sample scenario, and of course, it will drive the gradient to infinity.
- The PCA example is inconsistent with the optimal sparse representation definition, as its latent representation has no uncertainty at all.
- Reconstruction error is just an intuitive explanation of likelihood loss under the Gaussian observation model. For a proper theory of variational inference, the concepts should be instead defined by conditional likelihoods.
- Please use vector formats instead of bitmaps in the Figures.
- The experiments need to be substantially enriched.
- I do not feel the author(s) have shed real valuable insights to guide practice. What does the statement "this result suggests that heuristic modifications to or constraints on the VAE energy function may be ill-advised, and large gradients should be accommodated to the extent possible." even mean? Such confusing.

**Main Review:**

===== After Rebuttal =====

Please refer to my "After Rebuttal" post.

=====================

~~This is clearly not a NeurIPS level submission. I know it is quite frustrating to receive a rejection, so I have patiently read this manuscript until page 5, trying my best to prepare as much feedback as possible to help the author(s) improve this work, then I gave up. Its contents are simply not technically sound. I scrolled down to the Experiment section and found out it is not adequate as well, so here I am writing this review. The language issues in writing (sentences too long, grammar issues, bad expressions, etc.) I can tolerate them, but there are factual errors in this paper, so my evaluation is a firm no. The author(s) have some fundamental misunderstanding of variational inference models, as reflected in the definitions, analysis, and examples presented in this work. This work will be of limited value to the community. See more detailed comments in the limitation section below.~~

**Time Spent Reviewing:**

3

---

> ### Author Response · Authors · 2021-08-10
> **Response to Reviewer 69AE**
>
> While we sincerely appreciate the effort it takes to review technical and at times nuanced NeurIPS submission, we were admittedly somewhat disappointed that the reviewer did not actually finish reading our paper after page 5.  Based on comments in the *Main Review* section of the feedback form, the claim was made that our paper had too many grammatical issues, factual errors, as well as some fundamental misunderstanding of variational inference models.  We respectfully disagree with each of these claims.  However, the only specifics provided to support the above criticisms are listed in the *Limitations and Societal Impact* section.  To this end we address each concrete comment/criticism below, which we do not believe undercut the value of our contribution given the explanations provided.
>
> **Question:**  Infinite gradients or overfitting? It turned out the infinite gradient issue the author(s) heavily focused on is caused by letting the precision parameter of the Gaussian noise model be trainable, which always explodes to infinity. Well, it is well known this is simply the overfitting in the finite sample scenario, and of course, it will drive the gradient to infinity.
>
> **Response:**  The issue of infinite gradients is not simply reducible to overfitting in a finite sample scenario.  In fact, all of our results hold equally well to the infinite sample case whereby $n \rightarrow \infty$ in Eq. (3) such that the VAE loss is expressible as integration over the ground-truth probability measure $\mu_{gt}$ defined in Section 1.  Instead, the infinite gradient issue we address is a direct consequence of applying maximum likelihood to data lying on a low-dimensional manifold in a high-dimensional ambient space (e.g., natural images). See Figure 1 for an example.
>
> **Question:**  The PCA example is inconsistent with the optimal sparse representation definition, as its latent representation has no uncertainty at all.
>
> **Response:**  This is not actually true, as PCA is exactly consistent with our definition of an optimal sparse representation (see Definition 1).  A couple points can help to clarify this issue.  First, from a stochastic perspective, probabilistic PCA is formally a special case of the VAE when the decoder is restricted to being affine; see for example reference (Dai et al., 2018) cited in the submission.  And for data lying on a low-dimensional linear subspace, this model will produce optimal sparse representations (per our definition) when $\gamma \rightarrow 0$.  And secondly, even regular deterministic PCA (which is directly computable via a linear AE) is consistent with our Definition 1, which explicitly accommodates AE models (see lines 99-101 and 106-107).  And again, if the data lie on a low-dimensional linear subspace, then regular PCA is capable of producing an optimal sparse representation per our stated criteria for AEs.
>
> **Question:**  Reconstruction error is just an intuitive explanation of likelihood loss under the Gaussian observation model. For a proper theory of variational inference, the concepts should be instead defined by conditional likelihoods.
>
> **Response:**  It was unfortunately unclear to us precisely what the reviewer is referring to by a proper theory of variational inference.  But at least for the analysis purposes of our paper, the reconstruction error we define is determined  by the data-dependent term of the VAE loss (or the deterministic analogue of an AE).  From Eq. (4), this then involves both the negative log of the conditional decoder distribution $p_\theta(x|z)$ (excluding normalization and scaling factors) and the encoder posterior $q(z|x)$.  In terms of our goal of examining low-dimensional sparse structure, this definition is fully adequate.
>
> **Question:**  Please use vector formats instead of bitmaps in the Figures.
>
> **Response:**  While it is not difficult to change figure formats, we do not believe this to be a notable limitation of our work.
>
> **Question:**  The experiments need to be substantially enriched.
>
> **Response:**  As this submission is primarily of an analytical nature, rather than an algorithmic contribution, we would argue that the current experiments (both in the main paper and supplementary) are sufficient to convey the desired message.
>
> **Question:**  What does the statement "this result suggests that heuristic modifications to or constraints on the VAE energy function may be ill-advised, and large gradients should be accommodated to the extent possible." even mean?
>
> **Response:**  The sentence the reviewer refers to is merely a summarization point referring back to material described in Section 4, which begins on page 7.  However, because the reviewer conceded not reading most material after page 5 (except for scrolling through later experimental sections), this content was likely inadvertently missed.  In this regard then, we politely recommend checking Section 4 to infer our meaning and background context.

---

### Official Review · Reviewer_f5mJ · 2021-07-17

**Rating:** 6
**Confidence:** 4

**Summary:**

The paper investigates a specific work regime for VAE characterized by close to infinite gradients. The authors states that this phenomenon occurs for small values of a regularization power $\gamma$ and cannot be avoided, if one wants a perfect data reconstruction.


**Main Review:**

Pros:
- Clear analysis of presented phenomenon with a theorem, nice illustrations, profound investigation.

- Present explanation of a well-known effect during the VAE training.

Cons:
- The paper provides only one possible reason for large gradients. However, in practice it can be the case, that infinite gradients phenomenon occurs due to different reasons, so there is no guarantee that the paper finally solves the problem.

- Optimization of gamma refers to Bayesian problem statement, thus a full Bayesian discussion of the problem at hand can provide more intuition on what is going on, and how can we avoid the infinite gradient effect with a proper prior on parameters.

- Formula (6) – why the objective function has h in the first term, as usually we don’t use any h() for the training of AE?
Maybe there exists another criterion for AE learning, that doesn’t require assumptions of Theorem 4 to get the perfect reconstruction with at most r nonzero rows of Z (for example, we impose only some regularisation on z in the second term in (6)?)?
Under which conditions on data we get that $x^{(i)} = \mu_x(z^{(i),\theta})$ requires that $\|z_k\|_2>0$ for at least r<d rows?

- Does the infinite gradients phenomenon occur in other types of models? What makes this statement unique? What is with GMM or Kernel PCA or bayesian regression or PCA? E.g., in case of kernel PCA we can also require the data interpolation property. Will we observe the same problem?

- Assumption in formula (4) seems unrealistic given typically noisy settings for VAE and for statistics in general. It can lead to bad solutions making further arguments invalid. A discussion is needed.
Do we observe the infinite gradients phenomenon in case we do not require the interpolation property?

- The present work uses insights from
B. Dai and D. Wipf. Diagnosing and enhancing VAE models. In ICLR, 2018.
I would expect some discussion on the relation of the new results and the results from that paper.

- I understand that in case of sufficiently clean data, lying on the manifold, the interpolation property is a reasonable requirement. According to the authors, we have to tune gamma to achieve it. Are there any specific algorithms for tuning gamma, not just general advices?

- The optimization problem in (6) follows from the VAE optimization problem (3) only if sigma_z is constant and gamma is small. Theorem 4 are proved for the optimization problem (6). I am OK to transfer the conclusions of Theorem 4 to the VAE case when gamma is small. But the requirement of constant sigma_z is not obvious. What does happen if we model sigma_z with some neural network and tune its parameters? Will we observe the same effects? Maybe adaptive sigma_z will compensate the instability?

In general, the paper is interesting, but the assumptions sometimes look too narrow or results are not formally connected, I need more clarifications.

Minor comments:
- Line 103 – should it be minimal instead of maximal?

- Line 325 - "spare representations". Maybe "sparse"?

**Time Spent Reviewing:**

2.0

---

> ### Author Response · Authors · 2021-08-10
> **Response to Reviewer f5mJ**
>
> We appreciate the comments, which are addressed point-by-point below.
>
> **Question:**  The paper provides only one possible reason for large gradients. However, in practice it can be the case, that infinite gradients phenomenon occurs due to different reasons, so there is no guarantee that the paper finally solves the problem.
>
> **Response:**  Just to clarify, we are not really attempting to solve the problem of infinite gradients per se.  Rather, we are more concerned with explaining the underappreciated fact that such gradients are a natural, often unavoidable consequence of models designed to find optimal low-dimensional sparse structure in high-dimensional data.  That infinite gradients might arise in other unrelated contexts is out of our scope.
>
> **Question:**  Optimization of $\gamma$ refers to Bayesian problem statement, thus a full Bayesian discussion of the problem at hand can provide more intuition on what is going on, and how can we avoid the infinite gradient effect with a proper prior on parameters.
>
> **Response:**  The variance $\gamma$ is just a deterministic parameter like any other within a VAE model.  In fact, all VAE model parameters, from both the encoder and decoder, are generally treated as deterministic values to be learned via SGD, not as part of a fully Bayesian treatment whereby such unknowns are stochastic.  Beyond this, we are not aware of any alternative VAE formulation whereby a prior is applied to these parameters, and hence we have devoted our attention to the standard Gaussian VAE model as currently in common use.
>
> **Question:**  Formula (6) – why the objective function has $h$ in the first term, as usually we don’t use any $h()$ for the training of AE? Maybe there exists another criterion for AE learning, that doesn’t require assumptions of Theorem 4 to get the perfect reconstruction with at most r nonzero rows of $Z$ (for example, we impose only some regularisation on $z$ in the second term in (6)?)? Under which conditions on data we get that  requires that  for at least $r < d$ rows?
>
> **Response:**  Actually, we only use the function $h$ in the first term of Eq. (6) so as to more directly link this AE structure with VAE models (e.g., see Lemma 3).  However, in fact Eq. (6) and Theorem 4 can be generalized to replace $h$ in the first term with any function such that the global minimum of (6) produces zero reconstruction error.  Additionally, for $d$-dimensional data drawn in general position from an $r$-dimensional manifold, we will generally require that at least $r$ rows of $Z$ must be nonzero.
>
> **Question:**  Does the infinite gradients phenomenon occur in other types of models? What makes this statement unique? What is with GMM or Kernel PCA or bayesian regression or PCA? E.g., in case of kernel PCA we can also require the data interpolation property. Will we observe the same problem?
>
> **Response:**  Note that probabilistic PCA is just a special case of a VAE, which displays the exact same form of unbounded gradient phenomena; similarly for any model that attempts to maximize the likelihood of data lying on or near a low-dimensional manifold/subspace, e.g., natural images or related.
>
> **Question:** Assumption in formula (4) seems unrealistic given typically noisy settings for VAE and for statistics in general. It can lead to bad solutions making further arguments invalid. A discussion is needed. Do we observe the infinite gradients phenomenon in case we do not require the interpolation property?
>
> **Response:**  The assumption from Eq. (4) is made in part for analysis purposes, and in part because for many types of data lying on or at least very near low-dimensional manifolds (which is essentially all natural images), our definition of an optimal sparse representation directly aligns with the dimensionality of such manifolds.  And Definition 1 also corresponds with a long precedent in the signal and image processing literature involving the search for low-dimensional structure in high-dimensional data.  We can include additional supporting references to a revision, e.g., seminal papers such as "Optimally sparse representation in general (nonorthogonal) dictionaries via $\ell_1$ minimization," "Robust uncertainty principles: Exact signal reconstruction from highly incomplete frequency information," and "Exact matrix completion via convex optimization" among many others.
>
> Moreover, as mentioned in footnote 1 of our submission, we can in principle relax the requirement of perfect reconstructions to accommodate some degree of approximation error.  In this scenario, we envision that the basic intuitions elucidated in our paper would still apply, even if the exact assumptions of our main technical results do not strictly hold. This is common practice in analysis, whereby simplified noise-free conditions are assumed for tractability.
>
> **Question:**  The present work uses insights from B. Dai and D. Wipf. Diagnosing and enhancing VAE models. In ICLR, 2018. I would expect some discussion on the relation of the new results and the results from that paper.
>
> **Response:**  We have cited Dai and Wipf seven different times in our submission, with the relevant context in the text.  We are not sure exactly what additional discussion is needed to further clarify the relationship; however, we are very open to feedback in this regard.  Loosely speaking though, Dai and Wipf could be viewed as more focused on improving existing VAE models, while our present submission is related to understanding how certain VAE phenomena can translate to broader classes of models designed to find low-dimensional structure in data.
>
> **Question:**  I understand that in case of sufficiently clean data, lying on the manifold, the interpolation property is a reasonable requirement. According to the authors, we have to tune $\gamma$ to achieve it. Are there any specific algorithms for tuning $\gamma$, not just general advices?
>
> **Response:**  The variance $\gamma$ is a VAE decoder parameter that can be learned via SGD as with any other VAE parameter.  And as discussed in Section 4, this can be useful in the context of finding optimal sparse representations.
>
>
> **Question:**  The optimization problem in (6) follows from the VAE optimization problem (3) only if $\sigma_z$ is constant and $\gamma$ is small. Theorem 4 are proved for the optimization problem (6). I am OK to transfer the conclusions of Theorem 4 to the VAE case when $\gamma$ is small. But the requirement of constant $\sigma_z$ is not obvious. What does happen if we model $\sigma_z$ with some neural network and tune its parameters? Will we observe the same effects? Maybe adaptive $\sigma_z$ will compensate the instability?
>
> **Response:**  Lemma 2 and Lemma 3 detail special cases that link the optimization problem from Eq. (6) to VAE models.  For Lemma 2, the stated condition does involve setting $\sigma_z$ to a trainable constant $s$; however, this is not a significant restriction since the global optimum in this case is achievable via a constant $\sigma_z$ (note the equivalence described on lines 192-193).  In contrast, for Lemma 3 it is stipulated that $\sigma_z$ is pushed to zero.  This is natural given that generally speaking AE models are deterministic so it is necessary to constrain the variance of the VAE encoder to connect with a general AE.  Note also that it has already been established in prior work that VAEs with arbitrary $\sigma_z$ can have unbounded gradients.  The point here is merely to show that AE models designed to guarantee optimal sparse representations with general data must also have infinite gradients.
>
>
> **Question:**  Line 103 – should it be minimal instead of maximal?
>
> **Response:**  Maximal is actually correct, i.e., we want the most latent dimensions to be uninformative while still maintaining perfect reconstructions.
>
> **Question:**  Line 325 - "spare representations". Maybe "sparse"?
>
> **Response:**  Thanks for pointing this out, yes it should be sparse.

---

> > ### Comment · Reviewer_f5mJ · 2021-09-02
> > **Decided to increase the score to 6 after analysing all discussions**
> >
> > - Unfortunately, due to other obligations I was not able to take part in the discussions. I really appreciate the hard work done by other reviewers in discussing the paper with the authors.
> > - The authors clearly addressed my questions/comments
> > - I think that the main misunderstanding is due to the very broad claims initially made by the authors. If the authors introduced their setting (noise-free data on a low-dimensional manifold, gaussian model, requirements to optimal sparse representation) from the very beginning, it would be easier to understand the following results. However, the authors made some rather broad claims based on conclusions they got when analysing the model under the setting mentioned above. Fortunately, the authors confirmed they can re-write the paper in more appropriate way
> > - Anyway, the analysis, made in the paper, is useful and interesting, so I increased my score.

---

### Comment · Reviewer_69AE · 2021-08-11
**Why I consider the premises of this paper is wrong.**

First, I want to apologize for the tone in my original review, which is unpolite and aggressive. I would like to thank the author(s) for taking the time to carefully respond to my comment, which I have carefully read and will actively respond to.  Before I comment on the author's rebuttal, I would like to take a moment to elaborate on why I consider this work problematic. Specific responses to the technical details will unlikely change my overall evaluation of this paper.

Before presenting my argument, let us recap what this paper is about. This work is titled "On the Value of Infinite Gradients in Variational Autoencoder Models". The part "Variational Autoencoder Models" pulls me into this reviewer pool, for being a veteran on the topic. A recurring theme in this paper is the concept of "zero-reconstruction error" (e.g., Eq. (4)), which is used in the definition of optimal sparse representation (Definition 1). Well, this is where everything starts to fall apart from the perspective of variational inference.

To be clear, the concept of zero-reconstruction error does not even apply to variational inference models, as this means the conditional likelihood $p(x|z)$ degenerate to a Dirac delta measure. In other words, if the reconstruction is perfect, then there is not much point in introducing latent space uncertainty at all. That also implies your model has no generalization ability at all: the data space will be an immersion of the latent space, and you can find a map that exactly maps some latent to the observed data points. However, the mapping from the remaining latent space to those data points outside the empirical support of training data can be arbitrary. And I am not even mentioning how one gonna reconcile the fact $KL(q(z|x)||p(z))$ is gonna be degenerate as well. So I encourage the author(s) to brush up on their knowledge of variational inference by reading papers such as [Blei et al. (2017)]. The original VAE paper is also a must-read, as some of my points (e.g., generalization) are adequately discussed there.

I will be happy to reconsider my evaluation if this is just an autoencoder paper. However, if the author(s) are to keep the term "variational" in the title, I will not endorse a positive recommendation.

**References**
* [Blei et al. (2017)] Variational Inference: A review for Statisticians. JASA
* [Kingma et al. (2014)] Auto-encoding Variational Bayes. ICLR 2014

---

> ### Author Response · Authors · 2021-08-12
> **Why the premises of this paper are relevant**
>
> We appreciate the follow-up engagement.  The primary point of contention appears to be our definition of optimal sparse representations (Definition 1) as particularly applied to probabilistic VAEs, not deterministic AEs.  More specifically, the reviewer states that "the concept of zero-reconstruction error does not even apply to variational inference models."  However, we respectfully disagree with this claim.  In the *limit* as $\gamma \rightarrow 0$, variational inference is indeed often applied to producing perfect reconstructions of data, with many desirable properties related to finding low-dimensional structure.
>
> For example, even the variational review article from Blei et al. (2017) that the reviewer mentioned references work doing exactly this; please see the section called "Bayesian Linear Regression with Automatic Relevance Determination" and the reference "A New View of Automatic Relevance Determination" NIPS 2008 cited within.  With respect to the latter, a variational procedure in the limit of variance converging to zero is applied to solving the exact sparse reconstruction problem
>
> $\min_z || z ||_0 ~~ \mbox{s.t.}~ ||x - A z|| = 0,$
>
> where $A$ is an overcomplete design matrix, $x$ are observations, and $z$ are sparse coefficients to learn by minimizing the $\ell_0$ (pseudo) norm.  Moreover, even within the present context of VAEs, it has already been proven that for data lying on a low-dimensional manifold, a VAE with sufficient decoder capacity to model the manifold can perfectly reconstruct *any* sample drawn from within it in the limit as $\gamma \rightarrow 0$.  Please see Theorem 5 in reference Dai and Wipf (2019) of our submission.  Additionally, as a simple test one can train a VAE with an affine decoder on data drawn from a low-dimensional linear subspace. In this situation $\gamma$ as well as reconstruction errors will converge to zero, and the decoder weights will span the smallest principal subspace containing the data.  This then culminates in an optimal sparse representation per our definition.
>
> Of course the above obviously implies that the conditional likelihood $p(x|z)$ approaches a Dirac delta measure, and the factor $KL[q(z|x)||p(z)]$ diverges as the reviewer suggests.  But these limiting conditions are in fact quite suitable for recovering optimal low-dimensional structure, even if it comes at the cost of compromising generative modeling capabilities and well-behaved densities.  Actually, this is the main point of Dai and Wipf (2019); namely, for data lying on a low-dimensional manifold, a likelihood-based generative model like the VAE will favor perfect reconstructions over the accuracy of generated samples at global optima. And subsequently, a second-stage model is required to actually recover the correct distribution within the manifold once the manifold has been accurately estimated (via perfect reconstructions) by the first VAE stage.
>
> Regardless, the critical message is this:  VAEs have value beyond just generative modeling, as they can be applied as a form of regularized AE for finding optimal low-dimensional structure with various downstream applications as we discuss in Section 2.2.  And for data lying on a low-dimensional manifold (like most natural images), guaranteeing that such structure aligns with global optima will generally involve a loss function with unbounded gradients, which is one of the major points of our submission.
>
> Perhaps one way to better communicate these issues is to revise our original Definition 1 to consolidate the separate VAE and AE notions of optimal sparsity as follows:
>
> **Definition 1 (Revised/Simplified version)**
> >An autoencoder-based architecture (VAE or otherwise) with encoder component $\mu_z\left( \cdot; \phi  \right)$ and decoder $\mu_x\left( \cdot; \theta  \right)$ (with $\theta \in \Theta$) produces an **optimal sparse representation** of a training set $X$ if the following two conditions simultaneously hold:
> >
> >1) The reconstruction error is zero, meaning $\frac{1}{n} \sum_{i=1}^n || x^{(i)} - \mu_x [ \mu_z (x^{(i)}; \phi  ); \theta ] ||_2^2 = 0.$
> >
> >2) Conditioned on achieving perfect reconstructions per criteria (1) above, the number of latent dimensions such that $\mu_z\left(x^{(i)}; \phi  \right)_j  = 0$ for all $i$ is maximal across any $\theta \in \Theta$.
>
>
>
> The above definition is still aligned with Section 3 and Theorem 4, while avoiding any questions about variances converging to zero.  We may then connect back to VAEs by noting that, in the limit $\gamma \rightarrow 0$, an optimal VAE solution can satisfy the conditions of this Definition 1 when
>
> $\frac{1}{n} \sum_{i=1}^n E_{ q_{\phi}(z|x^{(i)}) }   ( ||x^{(i)} - \mu_x [ z; \theta ] ||_2^2 )   \rightarrow$
>
> $ \frac{1}{n} \sum_{i=1}^n || x^{(i)} - \mu_x [ \mu_z (x^{(i)}; \phi  ); \theta ] ||_2^2 = 0,$
>
> while the maximal number of latent dimensions converge to the prior, i.e., $q_{\phi}(z_j|x^{(i)} ) = N(0,1)$.  In contrast, for active dimensions, $\sigma_z (x^{(i)}; \phi  )_j \rightarrow 0$ allowing for arbitrarily accurate reconstructions.  And just to be clear, this claim is *not* a contribution of our paper, but rather a result from reference Dai and Wipf (2019) which has already been cited extensively for this specific purpose in various places (e.g., lines 118, 127, 133, and 199).  Anyway, we are happy to revise the paper to reflect this alternate version of Definition 1. Our main contributions and analysis in the main sections of the paper need not change to accommodate.

---

> > ### Comment · Reviewer_ATkA · 2021-08-18
> > **Optimal lossless sparse representation?**
> >
> > As you can see from my review I share some of the criticism with the goal of a (V)AE always being zero reconstruction error. In contrast to reviewer 69AE I think that there are use-cases in practice where (V)AEs are being used with the goal of having (near) zero reconstruction error, and broadly agree with the response laid out by the authors above.
> >
> > However I would be much happier if the definition above could be renamed as "optimal **lossless** sparse representation" or similar, since optimal sparse representations for tasks where zero-reconstruction error is undesirable would not fall under that definition. One could then also use e.g. rate-distortion theory to define "optimal lossy sparse representations" for the latter tasks/applications.

---

> > > ### Author Response · Authors · 2021-08-18
> > > **Optimal lossless sparse representation response**
> > >
> > > That's a fair point, and we definitely want to avoid any unnecessary ambiguity.  It's also an easy fix to make such a change, thanks for the suggestion.

---

### Comment · Reviewer_69AE · 2021-09-02
**After rebuttal & discussion**

First, I would like to apologize again for having acted a bit radical in my original review. That is unacceptable and unprofessional. I am keeping the original comment not to uphold it but as a minder to myself and everyone, that a constructive review should be composed courteously, even if one has a strong opinion against the content or presentation of a paper. But as some other reviewers also pointed out, some statements in the original draft have over-generalized user application settings, to the extent, they are too narrowly focused (i.e., lossless versus lossy compression). So that is the honest response from some readers if they would find the setting controversial. Maybe next time the author(s) could try first to get some constructive feedback from other knowledgeable colleagues that may have different opinions (e.g., in the same area but have a different background (in this case, maybe colleagues in EE, stats, math, physics)), because sometimes you just do not know what will strike a nerve.

Sorry due to more urgent obligations I was unable to further engage with the author(s) as promised at the beginning of this discussion. It has deprived the opportunity for both sides to further clarify their positions, which is unfortunate. But I have carefully read the exchanges between the author(s) and reviewer ATkA, and the references the author(s) pointed out in their response to my comments. Here are some follow-up comments.

1. I do recognize the significance of studying optimal reconstruction, but I remain unconvinced of its relevance in VAE settings. I reserve the right to disagree with some prior literature, even if they are published and highly cited (I believe I actually physically attended some of the presentations on those works or at least related work from the same author(s)). In practice, I concur there are many important applications optimal reconstruction is desired, but I do not see VAE as a principled approach to address this problem (the seminal works that the author(s) mentioned in the rebuttal such as "Optimally sparse representation in general (nonorthogonal) dictionaries via l1 minimization," "Robust uncertainty principles: Exact signal reconstruction from highly incomplete frequency information," and "Exact matrix completion via convex optimization" might be more relevant.) The "Bayesian Linear Regression with Automatic Relevance Determination" section in Blei (2017) and the reference "A New View of Automatic Relevance Determination" NIPS 2008 are not exactly what I would align with what is studied in this research. My major objections are listed in my post "Why I consider the premises of this paper is wrong", and I maintain they are valid criticisms. Now I see the author(s) are approaching from a different perspective, so I retract my claim this is a bad paper.

2. I am glad to see the concessions the author(s) have made to soften their claim and promised changes to their statement, as similar concerns are shared by many reviewers.

3. Overall I think the discussion is fruitful and constructive. I am adjusting my score to reflect an update of my understanding.

---

> ### Author Response · Authors · 2021-09-03
> **Closing response**
>
> While we sincerely appreciate the polite closing remarks and apologetic tone, it is our understanding that the rolling discussion period is now officially over and ACs have already made or are making decisions.  So while we would otherwise like to respond in more depth to the reviewer's new comments, this now seems unproductive.  That being said, at least for posterity we have already provided a comprehensive openreview response (see below) to the reviewer's earlier post "Why I consider the premises of this paper is wrong."  Additionally, we fully grant that the reviewer has the right to disagree with prior published work.  However, if this disagreement is to form the basis for downgrading the present submission, then we would respectfully argue that the burden of proof is on the reviewer to actually provide concrete support or evidence for why the prior work in question is somehow flawed or misplaced.

---

### Decision · Program_Chairs · 2021-09-28

**Decision:**

Accept (Spotlight)

**Comment:**

The authors show that infinite gradients are required to recover optimally sparse representations of data in autoencoder models using a Gaussian likelihood with a learned variance.

The results in the paper appear both correct and interesting in the context of autoencoders, but the reviewers found some of the claims insufficiently precise and at times overstated, which is a serious issue for a theoretical work. Most notably, they were not convinced by the claims of the applicability of the results to VAEs (e.g. the zero reconstruction error as one of the assumptions). The back-and-forth with the reviewers provided some necessary clarifications and explanations, and generated some excellent suggestions for improving the paper. Unfortunately implementing the changes required would amount to a major revision and thus is not feasible without resubmission. The paper would also really benefit from a section covering prior work on infinite gradients, including a detailed discussion of the relevant results in Dai and Wipf (2019) and their relationship to those in this paper.

**Consistency Experiment:**

NeurIPS has a long history of experimentation. In 2014, NeurIPS ran an experiment in which 10% of submissions were reviewed by two independent committees to quantify the randomness in the review process. This year, we repeated a variant of this experiment to see how the quality of the review process has changed over time.  This paper was part of the experiment and was therefore assigned to two committees (consisting of reviewers, an Area Chair, and a Senior Area Chair) that reached independent decisions.  If both committees made the same recommendation, this recommendation was followed. If a single committee recommended acceptance, the paper was accepted (with the exception of a few cases in which the other committee identified what we considered a fatal flaw, e.g., an error in a key result).

This copy’s committee reached the following decision: **Reject**

The other committee assigned to the paper recommended **Accept (Spotlight)**.  You can find the other set of reviews, along with any follow up discussion with the authors here:
https://openreview.net/forum?id=oumDUrf2dAB